# From Low to High-Value Designs: Offline Optimization via Generalized Diffusion

## Abstract

This paper studies the black-box optimization task which aims to find the maxima of a black-box function using only a static set of its observed input-output data. This is often achieved via learning and optimizing a surrogate function using such offline dataset. Alternatively, it can also be framed as an inverse modeling task which maps a desired performance to potential input candidates that achieve it. Both approaches are limited by the limited amount of offline data. To mitigate this limitation, we introduce a new perspective which cast offline optimization as a diffusion process mapping between an implicit distribution of low-value inputs (i.e., offline data) and a superior distribution of high-value inputs (i.e., solution candidates). Such diffusion process can be learned using low- and high-value inputs sampled from synthetic functions that resemble the target function. These synthetic functions are constructed as the mean posterior of multiple Gaussian processes fitted with different parameterizations on the offline data, alleviating the data bottleneck. Experimental results demonstrate that our approach consistently outperforms previous methods, establishing a new state-of-the-art performance.

## 1 Introduction

Numerous engineering and science applications across various domains involve optimization tasks over complex design spaces guided by running physical lab experiments or computer simulations (Wang et al., 2023). For example, designing energy-efficient hardware accelerators (Ceze et al., 2016; Bogdan et al., 2019; Kim et al., 2018) for emerging applications/domains requires expensive cycle-accurate simulations to evaluate hardware configurations. Searching for nanoporous materials with high adsorption property for absorbing $CO_2$ from the air or storing hydrogen gas for hydrogen-powered vehicles requires expensive lab experiments for evaluation (Deshwal et al., 2021; Gantzler et al., 2023). Other examples include designing proteins (Gao et al., 2020), molecules (Deshwal & Doppa, 2021), and drugs (Schneider et al., 2020) via running numerous experiments.

**Black-Box Optimization.** As these experiments are often costly, labor-intensive, and have expensive overhead (e.g., the cost of procuring materials and equipment), it is impractical for domain practitioners to adopt online optimization techniques, such as Bayesian optimization (BO) (Snoek et al., 2012; 2015; Wang et al., 2018), which rely on running iterative experiments to find the best design. A more realistic solution approach (Brookes et al., 2019; Kumar & Levine, 2020b) aims to maximize a black-box experimental process via learning and optimizing a surrogate model using an existing dataset of its input-output queries.

**Challenges.** The main problem with this paradigm is that the surrogate's predictions can become increasingly erroneous when the search moves away from the offline dataset, especially when the surrogate model overfits on biased and/or sparse training data.

**Prior Approaches.** To mitigate this issue, most existing approaches have focused on advancing techniques in (1) **forward modeling** that penalize high-value surrogate predictions at out-of-distribution (OOD) inputs (Trabucco et al., 2021; Chen et al., 2024; Yuan et al., 2023), (2) **inverse modeling** that find the most promising regions of (reliable) high-performing inputs (Kumar & Levine, 2020a; Nguyen et al., 2023; Krishnamoorthy et al., 2022; 2023; Chemingui et al., 2024) to sidestep the OOD issue of forward modeling, and (3) **search policies** that learn a direct plan to navigate from low-value inputs to high-value inputs (Krishnamoorthy et al., 2022; Chemingui et al., 2024).

**Limitations.** Despite their promising results, forward and inverse approaches depend on learning a mapping (or inverse mapping) between input designs and their corresponding performance outputs using offline data. As a result, their effectiveness is inherently limited by the availability of data. Likewise, learning direct search policies also suffers from the same data bottleneck since these methods still need to sample heuristic trajectories from the offline dataset to use as learning feedback.

Furthermore, information regarding regions with high-performing inputs is often not observable from the offline data, especially in low-data scenarios which might further restrict the effectiveness of the learned models/policies. To mitigate such data bottleneck, we propose to approach offline optimization from a new perspective of distributional translation, as highlighted below.

**New Perspective.** The offline dataset is modeled as an implicit distribution of low-value designs, and offline optimization is recast as learning a probabilistic transformation that translates it into a (better) distribution of high-value inputs. Interestingly, although the feedback needed to learn this transformation is absent in the offline dataset, it can be derived from a distribution of synthetic functions that are similar to the (unknown) target function (up to a scale factor). As these functions can be provably constructed across various output scales, alleviating the data bottleneck and consequently, broaden the solution scope of offline optimization.

**Technical Contributions.** This is substantiated via the followings:

**1.** A diffusion-based framework for offline optimization which studies a direct parameterization of the high-value input distribution in terms of the output of a multi-step surrogate-based gradient ascent. This reveals a correspondence between offline optimization and a generalized diffusion process (Ho et al., 2020) mapping between the (implicit) distribution of (low-value) offline input and another (target) input distribution. The resulting diffusion process bypasses the explicit construction of a surrogate model, allowing for external (surrogate-free) guiding information to be used for model training, alleviating the data bottleneck of offline optimization (Section 3.1).

**2.** A practical pre-training and adaptation framework that (1) learns multiple Gaussian process priors (Williams & Rasmussen, 2006) over synthetic functions similar to the target function and (2) samples low- and high-value inputs from their corresponding mean functions to construct and pre-train our generalized diffusion model. The pre-trained diffusion model can be used to map from the offline input distribution to a better distribution over inputs with higher performance (Section 3.2).

**3.** An extensive empirical evaluation on a variety of benchmark datasets (Trabucco et al., 2022) establishing a new state-of-the-art performance, which significantly and consistently improves over previous work. Our empirical evaluation also features a rich ablation studies examining in detail the practical impact of different components of our framework on its performance, as well as its sensitivity to several key hyperparameters (Section 4).

For clarity, we also provide a concise related work and background review on Gaussian processes (Rasmussen & Williams, 2006) and diffusion models (Ho et al., 2020) in Sections 2 and 5.

## 2 PROBLEM DEFINITION AND BACKGROUND

We formally define the offline optimization task in Section 2.1 and provide concise background review on Gaussian processes (Williams & Rasmussen, 2006) and diffusion models (Sohl-Dickstein et al., 2015; Ho et al., 2020) in Sections 2.2 and 2.3, respectively.

### 2.1 PROBLEM DEFINITION

Offline optimization is formulated as the maximization of a black-box function $g(\boldsymbol{x})$ using only an offline dataset of observations $D = \{(\boldsymbol{x}_i, y_i)\}_{i=1}^n$ which denote the past input-output queries extracted from $g(\mathbf{x})$ in previous experiments. A direct approach to this problem is to learn a surrogate $g(\boldsymbol{x}; \boldsymbol{\omega}_*)$ of $g(\boldsymbol{x})$ via fitting its parameter $\boldsymbol{\omega}_*$ to the offline dataset,

$$\boldsymbol{\omega}_* \quad \triangleq \quad \arg\min_{\boldsymbol{\omega}} L(\boldsymbol{\omega}) \quad \triangleq \quad \arg\min_{\boldsymbol{\omega}} \sum_{i=1}^n \ell\Big(g(\boldsymbol{x}_i; \boldsymbol{\omega}), \, y_i\Big) , \tag{1}$$

where $\boldsymbol{\omega}$ denotes a parameter candidate of the surrogate and $\ell(g(\boldsymbol{x}; \boldsymbol{\omega}), y)$ denotes the prediction loss of $g(.; \boldsymbol{\omega})$ on $\mathbf{x}$ if its oracle output is $y$. The (oracle) maxima of $g(\mathbf{x})$ is then approximated via,

$$\boldsymbol{x}_* \quad \triangleq \quad \arg\max_{\boldsymbol{x}} \ g(\boldsymbol{x}; \boldsymbol{\omega}_*) . \tag{2}$$

The main issue with this approach is that $g(\boldsymbol{x}; \boldsymbol{\omega}_*)$ often predicts erratically at out-of-distribution (OOD) inputs. To mitigate this, numerous surrogate or search regularizers have been proposed to either penalize the high-value surrogate prediction at OOD inputs (Trabucco et al., 2021; Chen et al., 2024; Yuan et al., 2023) or find an inverse mapping from the desired output to potential inputs (Krishnamoorthy et al., 2023; Nguyen et al., 2023), as further detailed in Section 5.

## 2.2 Gaussian Processes

A Gaussian process (GP) (Rasmussen & Williams, 2006) defines a probabilistic prior over a random function $g(\boldsymbol{x})$. It is parameterized by a mean function $m(\boldsymbol{x}) = 0$[1] and a kernel function $k(\boldsymbol{x}, \boldsymbol{x}')$. These functions induce a marginal Gaussian prior over the evaluations $\boldsymbol{g} = [g(\boldsymbol{x}_1) \ldots g(\boldsymbol{x}_n)]^\top$ of any finite subset of inputs $\{\boldsymbol{x}_1, \ldots, \boldsymbol{x}_n\}$. Let $\boldsymbol{x}_\tau$ be an unseen input whose corresponding output $g_\tau = g(\boldsymbol{x}_\tau)$ we wish to predict. The Gaussian prior over $[g(\boldsymbol{x}_1) \ldots g(\boldsymbol{x}_n) \, g(\boldsymbol{x}_\tau)]^\top$ implies:

$$g_\tau \quad \triangleq \quad g(\boldsymbol{x}_\tau) \mid \boldsymbol{g} \quad \sim \quad \mathbb{N}\Big(\boldsymbol{k}_\tau^\top \boldsymbol{K}^{-1} \boldsymbol{g}, \ k(\boldsymbol{x}_\tau, \boldsymbol{x}_\tau) - \boldsymbol{k}_\tau^\top \boldsymbol{K}^{-1} \boldsymbol{k}_\tau\Big) , \tag{3}$$

where $\boldsymbol{k}_\tau = [k(\boldsymbol{x}_\tau, \boldsymbol{x}_1) \ldots k(\boldsymbol{x}_\tau, \boldsymbol{x}_n)]^\top$ and $\boldsymbol{K}$ denotes the Gram matrix induced on $\{\boldsymbol{x}_1, \ldots, \boldsymbol{x}_n\}$ for which $\boldsymbol{K}_{ij} = k(\boldsymbol{x}_i, \boldsymbol{x}_j)$. Assuming a Gaussian likelihood $y \sim \mathbb{N}(g(\boldsymbol{x}), \sigma^2)$, it follows that

$$g_\tau \quad \triangleq \quad g(\boldsymbol{x}_\tau) \mid \boldsymbol{y} \quad \sim \quad \mathbb{N}\Big(\boldsymbol{k}_\tau^\top (\boldsymbol{K} + \sigma^2 \boldsymbol{I})^{-1} \boldsymbol{y}, \ k(\boldsymbol{x}_\tau, \boldsymbol{x}_\tau) - \boldsymbol{k}_\tau^\top (\boldsymbol{K} + \sigma^2 \boldsymbol{I})^{-1} \boldsymbol{k}_\tau\Big) , \tag{4}$$

which explicitly forms the predictive distribution of a Gaussian process. The defining parameter $\phi$ of $k(\boldsymbol{x}, \boldsymbol{x}')$ is crucial to the predictive performance and needs to be optimized via minimizing:

$$\ell(\boldsymbol{\phi}) \quad = \quad \frac{1}{2} \log \Big| \boldsymbol{K}_\phi + \sigma^2 \boldsymbol{I} \Big| + \frac{1}{2} \boldsymbol{y}^\top \Big( \boldsymbol{K}_\phi + \sigma^2 \boldsymbol{I} \Big)^{-1} \boldsymbol{y} , \tag{5}$$

where we now use the subscript $\phi$ to indicate that $\boldsymbol{K}$ is a function of $\phi$. In the context of this paper, we adopt the commonly used RBF kernel, $k(\boldsymbol{x}, \boldsymbol{x}') = \sigma^2 \exp(-0.5 \cdot \|\boldsymbol{x} - \boldsymbol{x}'\|^2 / \ell^2)$. Its kernel parameters are $\boldsymbol{\phi} = \{\sigma, \ell\}$ where $\sigma^2$ represents the signal variance, controlling the amplitude of the function, $\ell$ is the unit length-scale, governing the smoothness of the function.

## 2.3 Denoising Diffusion Probabilistic Model (DDPM)

Denoising diffusion probabilistic model (DDPM) (Sohl-Dickstein et al., 2015; Ho et al., 2020) characterizes (1) a forward diffusion process which map from an (implicit) data distribution to a isotropic Gaussian distribution; and (2) a parameterized backward diffusion process that reverses the forward process. In DDPM, the forward process keeps adding Gaussian noises to the input data until the distribution of the resulting noise-corrupted data converges to an isotropic Gaussian. This is achieved via simulating the following Markov chain starting from a data point $\boldsymbol{x}_0 \sim q_D(\boldsymbol{x})$,

$$q\Big(\boldsymbol{x}_1, \boldsymbol{x}_2, \ldots, \boldsymbol{x}_T \mid \boldsymbol{x}_0\Big) \quad = \quad \prod_{t=1}^{T} q\Big(\boldsymbol{x}_t \mid \boldsymbol{x}_{t-1}\Big) = \prod_{t=1}^{T} \mathbb{N}\Big(\boldsymbol{x}_t; \big(1 - \beta_t\big)^{1/2} \boldsymbol{x}_{t-1}, \beta_t \boldsymbol{I}\Big) , \tag{6}$$

with transition kernel $q(\boldsymbol{x}_t | \boldsymbol{x}_{t-1}) \triangleq \mathbb{N}(\boldsymbol{x}_t; (1 - \beta_t)^{1/2} \boldsymbol{x}_{t-1}, \beta_t \boldsymbol{I})$, where $\beta_t > 0$ is a small, manually set constant. This implies $q(\boldsymbol{x}_t \mid \boldsymbol{x}_0) = \mathbb{N}(\boldsymbol{x}_t; \boldsymbol{x}_0 \cdot \sqrt{\tilde{\alpha}_t}, (1 - \tilde{\alpha}_t) \boldsymbol{I})$ where $\tilde{\alpha}_t = (1 - \beta_1) \cdot (1 - \beta_2) \ldots (1 - \beta_t) < 1$. As such, when $t \to \infty$, $q(\boldsymbol{x}_t \mid \boldsymbol{x}_0) = \mathbb{N}(0, \boldsymbol{I})$. On the other hand, the reverse process aims to map from the isotropic distribution back to the original (implicit) data distribution via simulating another parameterized Markov chain,

$$p_\theta\Big(\boldsymbol{x}_0, \ldots, \boldsymbol{x}_{T-1} \mid \boldsymbol{x}_T\Big) \quad = \quad \prod_{t=1}^{T} p_\theta\Big(\boldsymbol{x}_{t-1} \mid \boldsymbol{x}_t\Big) \quad = \quad \prod_{t=1}^{T} \mathbb{N}\Big(\boldsymbol{x}_{t-1}; \boldsymbol{\mu}_\theta(\boldsymbol{x}_t, t), \beta_t \boldsymbol{I}\Big), \tag{7}$$

---

[1] We assume a zero mean function since the output can always be re-centered around 0.

Figure 1: The overall workflow of **L2HD**: (1) multiple Gaussian process posteriors are fitted on the offline data; (2) low- and high-value inputs of the resulting posterior mean functions were sampled and used to pre-train our generalized diffusion model which can map between different two (implicit) data distribution; and (3) the backward process of the pre-trained diffusion model is used on the target (offline) data to find high-performing inputs for the (unknown) target function.

starting from the isotropic Gaussian noise $\boldsymbol{x}_T \sim \mathbb{N}(0, \boldsymbol{I})$. The model is trained via maximizing a variational lower-bound $\text{ELBO}(\theta)$ of $\log p_\theta(\boldsymbol{x}_0)$ of the induced data marginal likelihood,

$$\text{ELBO}\Big(\theta\Big) \propto \sum_{t=2}^{T} D_{\text{KL}}\Big(q\Big(\boldsymbol{x}_{t-1} \mid \boldsymbol{x}_t, \boldsymbol{x}_0\Big)\|p_\theta\Big(\boldsymbol{x}_{t-1} \mid \boldsymbol{x}_t\Big)\Big) + \mathbb{E}_{q(\boldsymbol{x}_1|\boldsymbol{x}_0)}\Big[\log p_\theta\Big(\boldsymbol{x}_0 \mid \boldsymbol{x}_1\Big)\Big] , \quad (8)$$

where the conditional backward transition $q(\boldsymbol{x}_{t-1} \mid \boldsymbol{x}_t, \boldsymbol{x}_0) = \mathbb{N}(\boldsymbol{x}_{t-1}; \tilde{\boldsymbol{\mu}}(\boldsymbol{x}_t, \boldsymbol{x}_0), \tilde{\beta}_t \boldsymbol{I})$ can be derived in closed-form using Eq. 6. Its specific form is given via

$$\tilde{\boldsymbol{\mu}}\Big(\boldsymbol{x}_t, \boldsymbol{x}_0\Big) = \frac{1}{\sqrt{1-\beta_t}}\left(\boldsymbol{x}_t - \frac{\beta_t}{\sqrt{1-\tilde{\alpha}_t}} \cdot \boldsymbol{\epsilon}_t\right) \quad \text{and} \quad \tilde{\beta}_t = \left(\frac{1-\tilde{\alpha}_{t-1}}{1-\tilde{\alpha}_t}\right) \cdot \beta_t \quad (9)$$

where $\boldsymbol{\epsilon}_t \sim \mathbb{N}(0, \boldsymbol{I})$. Following Eq. 9, we can likewise parameterize the mean function of the backward transition model $\boldsymbol{\mu}_\theta(\boldsymbol{x}_t, t) \triangleq (1-\beta_t)^{-1/2} \cdot (\boldsymbol{x}_t - \beta_t \cdot (1-\tilde{\alpha}_t)^{-1/2} \cdot \boldsymbol{\epsilon}_\theta(\boldsymbol{x}_t, t))$. This consequently allows a simplification of the maximization task in Eq. 8 as

$$\theta \triangleq \underset{\theta}{\arg\min} \ \mathbb{E}_{\boldsymbol{x}_0, \boldsymbol{\epsilon}_t, t}\left[\left\|\boldsymbol{\epsilon}_\theta\left(\tilde{\alpha}_t^{1/2} \cdot \boldsymbol{x}_0 + \left(1-\tilde{\alpha}_t\right)^{1/2} \cdot \boldsymbol{\epsilon}_t, t\right) - \boldsymbol{\epsilon}_t\right\|^2\right] , \quad (10)$$

where $\boldsymbol{x}_0 \sim q_D(\boldsymbol{x})$, $\boldsymbol{\epsilon}_t \sim \mathbb{N}(0, \boldsymbol{I})$, and $t \sim [1, 2, \ldots, T]$. Once trained, we can use the learned backward process $p_\theta$ to decode from an isotropic Gaussian noise pattern $\boldsymbol{x}_T \sim \mathbb{N}(0, \boldsymbol{I})$ back to an authentic data point $\boldsymbol{x}_0 \sim q_D(\boldsymbol{x})$. Eq. 10 can also be generalized to include output information,

$$\theta \triangleq \underset{\theta}{\arg\min} \ \mathbb{E}_{(\boldsymbol{x}_0, y), \gamma, \boldsymbol{\epsilon}_t, t}\left\|\boldsymbol{\epsilon}_\theta\left(\tilde{\alpha}_t^{1/2} \cdot \boldsymbol{x}_0 + \left(1-\tilde{\alpha}_t\right)^{1/2} \cdot \boldsymbol{\epsilon}_t, (1-\gamma) \cdot y + \gamma \cdot y_\emptyset, t\right) - \boldsymbol{\epsilon}_t\right\|^2 \quad (11)$$

where $(\boldsymbol{x}_0, y) \sim q_D(\boldsymbol{x}, y)$, $\gamma \sim \text{Ber}(\kappa)$ is a Bernoulli random variable – with bias $\kappa \in (0, 1)$ – that determines whether (on this instance) we condition on the actual label $y$ or the default label $y_\emptyset = 0$. This is known as classifier-free diffusion guidance (Ho & Salimans, 2022).

## 3 OFFLINE OPTIMIZATION VIA DISTRIBUTIONAL TRANSLATION

This section presents a principled approach to learn a generalized diffusion process mapping from the (implicit) distribution over offline data to a superior distribution over the high-value input regimes. To achieve this, we parameterize such high-value input distribution as an affine transformation of an isotropic multivariate Gaussian noise. The transformation is in turn parameterized using the output of a multi-step surrogate-based gradient ascent. This reveals a generalized Brownian bridge diffusion process that transforms the offline data into samples from a distribution over high-value inputs (see Section 3.1). Learning this diffusion process therefore presents an alternative to black-box optimization. This can be achieved via sampling low- and high-value inputs from synthetic functions that resemble the target function. Such functions can be constructed as the posterior means of multiple Gaussian processes fitted with different parameter initializations on the offline data (see Section 3.2). See the overall workflow of our framework in Fig. 1.

### 3.1 Distributional Translation via Generalized Diffusion

First, we recall the conventional surrogate-based gradient ascent approach to offline optimization using a (learnable) surrogate $g(\boldsymbol{x}; \boldsymbol{\omega})$ (see Eq. 1),

$$\boldsymbol{x}_t \;=\; \boldsymbol{x}_{t+1} \;+\; \eta \cdot \nabla_{\boldsymbol{x}} g(\boldsymbol{x}_{t+1}; \boldsymbol{\omega}) \;\Rightarrow\; \boldsymbol{x}_t \;=\; \boldsymbol{x}_T \;+\; \eta \cdot \sum_{i=T}^{t+1} \nabla_{\boldsymbol{x}} g(\boldsymbol{x}_i; \boldsymbol{\omega}) \,, \tag{12}$$

which maps from a (offline) low-value input $\boldsymbol{x}_T$ to an input $\boldsymbol{x}_t$ with higher output value according to the surrogate $g(\boldsymbol{x}_i; \boldsymbol{\omega})$. Due to the distribution gap between the (implicit) offline data and the oracle distribution of high-value inputs $p(\widehat{\boldsymbol{x}}_0)$, $\boldsymbol{x}_t$ may correspond to an out-of-distribution (OOD) sample with respect to this distribution. This will result in poor offline optimization performance. To elaborate, as the oracle is a GP-distributed function, its high-value input $\widehat{\boldsymbol{x}}_0$ obtained via

$$\widehat{\boldsymbol{x}}_0 \;=\; \widehat{\boldsymbol{x}}_T \;+\; \eta \cdot \sum_{i=T}^{1} \nabla_{\widehat{\boldsymbol{x}}} g(\widehat{\boldsymbol{x}}_i)$$

is a random variable. Here, $\widehat{\boldsymbol{x}}_T = \boldsymbol{x}_T$ (in the offline data) but $\widehat{\boldsymbol{x}}_i$ and $\boldsymbol{x}_i$ are different iterations following (different) gradients from the oracle and surrogate, respectively.

**I. Black-Box Optimization via Sampling from Brownian Bridge Diffusion.** Our goal is to learn and sample from the oracle's high-value input distribution $p(\widehat{\boldsymbol{x}}_0)$ to solve offline optimization. This is however non-trivial since we do not know the oracle's gradient at unseen input and simulating this using the surrogate will cause a distributional gap. To compensate for this gap, we instead re-parameterize $p(\widehat{\boldsymbol{x}}_0)$ via viewing the random variable $\widehat{\boldsymbol{x}}_0 - \boldsymbol{x}_T$ as a transformation of an isotropic Gaussian noise $\boldsymbol{\epsilon}_t$ with transformation parameters $a_t \cdot (\boldsymbol{x}_t - \boldsymbol{x}_T)$ and $b_t$ as detailed below:

$$\widehat{\boldsymbol{x}}_0 \;-\; \boldsymbol{x}_T \;=\; a_t \cdot (\boldsymbol{x}_t \;-\; \boldsymbol{x}_T) \;+\; b_t \cdot \boldsymbol{\epsilon}_t \,, \tag{13}$$

where $a_t, b_t \in \mathbb{R}$ and $\boldsymbol{\epsilon}_t \sim \mathbb{N}(0, \boldsymbol{I})$ denote the isotropic multivariate Gaussian noise. This essentially re-parameterizes $p(\widehat{\boldsymbol{x}}_0)$ as an affine transformation of a Gaussian noise with parameters $b_t$ and $a_t \cdot (\boldsymbol{x}_t - \boldsymbol{x}_T)$ where $\boldsymbol{x}_t$ is determined in Eq. 12. Such parameterized transformation of Gaussian noise has often been used in flow-based models (Kobyzev et al., 2019) with various designs for the transformation parameters. Please note that $\widehat{\boldsymbol{x}}_0 \neq \boldsymbol{x}_0$ since their respective computation uses gradients from the surrogate and oracle, which are different. As such, this does not over-constrain the output of the surrogate-based gradient ascent in Eq. 12, which does not result in any contradictions.

Eq. 13 in turn surfaces a diffusion process whose transition can be learned to map between (implicit) distributions of the offline input and the oracle's high-value input in what follows. Choosing $a_t = 1/\zeta_t$ and $b_t = \delta_t^{1/2}/\zeta_t$ in Eq. 13,

$$\widehat{\boldsymbol{x}}_0 \;-\; \boldsymbol{x}_T \;=\; \frac{1}{\zeta_t} \cdot (\boldsymbol{x}_t \;-\; \boldsymbol{x}_T) \;+\; \frac{1}{\zeta_t} \cdot \sqrt{\delta_t} \cdot \boldsymbol{\epsilon}_t, \tag{14}$$

where $\zeta_t, \delta_t \in (0, 1)$. This inspires an alternate, more robust characterization of $\boldsymbol{x}_t$ in which it is re-parameterized as a random variable rather than a fixed quantity in Eq. 12,

$$\boldsymbol{x}_t \;\triangleq\; (1 - \zeta_t) \cdot \boldsymbol{x}_T \;+\; \zeta_t \cdot \widehat{\boldsymbol{x}}_0 \;-\; \sqrt{\delta_t} \cdot \boldsymbol{\epsilon}_t \,, \tag{15}$$

which can help bridge the distributional gap between $\boldsymbol{x}_0$ and $\widehat{\boldsymbol{x}}_0$, thus solving the offline optimization task more robustly. To elaborate, Eq. 12 involves a surrogate which is learned to fit only the offline data while Eq. 15 corresponds to a Brownian bridge diffusion process will be learned to map between the distribution of offline data and the high-value input distribution of the oracle. Replacing the former with the latter will help bridge the aforementioned distributional gap between $\boldsymbol{x}_0$ and $\widehat{\boldsymbol{x}}_0$.

To achieve this, we replace Eq. 12 with Eq. 15 above which in turn reduces to a Brownian bridge diffusion model (BBDM) (Li et al., 2023) when we choose $\zeta_t = 1 - m_t = 1 - t/T$ and $\delta_t = 2(m_t - m_t)^2$. It can then be used to bypass the need to learn an explicit surrogate in Eq. 12 that might drift away from the high-value input distribution. Instead, Eq. 15 now characterizes a diffusion process mapping between the offline and high-value input distributions, which no longer requires surrogate gradient to simulate $\boldsymbol{x}_t$. Learning the diffusion in Eq. 15 however requires sampling from the high-value regime of the oracle which is not possible. But, fortunately, we can sample instead

from the high-value regimes of similar functions to the oracle to provide surrogates for the oracle's high-value regime. This is discussed in detail in Section 3.2.

**II. Learning BBDM.** Solving offline optimization is then equivalent to learning this BBDM and then simulating samples from it. To learn this BBDM, we leverage its properties as detailed in Appendix C.2 to establish a Gaussian form for the reversed forward conditional $q(\boldsymbol{x}_{t-1} \mid \boldsymbol{x}_t, \widehat{\boldsymbol{x}}_0, \boldsymbol{x}_T)$,

$$q\Big(\boldsymbol{x}_{t-1} \mid \boldsymbol{x}_t, \widehat{\boldsymbol{x}}_0, \boldsymbol{x}_T\Big) \;\; = \;\; \mathbb{N}\Big(\boldsymbol{x}_{t-1}; \tilde{\boldsymbol{\mu}}\Big(\boldsymbol{x}_t, \widehat{\boldsymbol{x}}_0, \boldsymbol{x}_T\Big), \tilde{\delta}_t \cdot \boldsymbol{I}\Big), \tag{16}$$

where the detailed derivation of the mean and variance in Eq. 16 are deferred to Appendix C.2. Following Eq. 8, we can learn the BBDM via maximizing a variational lower-bound $\mathrm{ELBO}(\theta)$ of the marginal backward process $\log p_\theta(\widehat{\boldsymbol{x}}_0)$,

$$\begin{aligned} \mathrm{ELBO}(\theta) \;\; \propto \;\; & \sum_{t=2}^{T} D_{\mathrm{KL}}\Big(q\Big(\boldsymbol{x}_{t-1} \mid \boldsymbol{x}_t, \widehat{\boldsymbol{x}}_0, \boldsymbol{x}_T\Big) \| p_\theta\Big(\boldsymbol{x}_{t-1} \mid \boldsymbol{x}_t, \boldsymbol{x}_T\Big)\Big) \\ & + \;\; \mathbb{E}_{q(\boldsymbol{x}_1 \mid \widehat{\boldsymbol{x}}_0)}\Big[\log p_\theta\Big(\widehat{\boldsymbol{x}}_0 \mid \boldsymbol{x}_1, \boldsymbol{x}_T\Big)\Big], \end{aligned} \tag{17}$$

where we parameterize the backward process $p_\theta(\boldsymbol{x}_{t-1} \mid \boldsymbol{x}_t, \boldsymbol{x}_T) \triangleq \mathbb{N}(\boldsymbol{x}_{t-1}; \boldsymbol{\mu}_\theta(\boldsymbol{x}_t, \boldsymbol{x}_T, t), \tilde{\delta}_t \cdot \boldsymbol{I})$ with $\boldsymbol{\mu}_\theta(\boldsymbol{x}_t, \boldsymbol{x}_T, t)$ is obtained from replacing $\boldsymbol{\epsilon}_t$ in $\tilde{\boldsymbol{\mu}}(\boldsymbol{x}_t, \widehat{\boldsymbol{x}}_0, \boldsymbol{x}_T)$ with a parameterized noise prediction network $\boldsymbol{\epsilon}_\theta(\boldsymbol{x}_t, t)$. To enable the optimization of Eq. 17, we need to obtain the guiding information $\widehat{\boldsymbol{x}}_0$, e.g., an example of a high-performing input from a similar function. This is achieved via constructing a prior over a wide spectrum of artificial functions that are guaranteed to be similar to the target function. Low- and high-value input designs of the sampled function can then be used as samples of $(\boldsymbol{x}_T, \widehat{\boldsymbol{x}}_0)$ to learn this Brownian bridge diffusion model, as detailed next.

**Remark.** We are using the reverse index where $\widehat{\boldsymbol{x}}_T = \boldsymbol{x}_T$ denotes the low-value input design while $\widehat{\boldsymbol{x}}_0$ denotes the high-value input design for notational convenience. This is because we are drawing correspondence between the augmented gradient ascent and the marginal reverse conditional of the forward diffusion $q(\boldsymbol{x}_t \mid \boldsymbol{x}_{t-1})$ and the forward diffusion is supposed to go from high-value to low-value distributions, which means going from $\boldsymbol{x}_{t-1}$ to $\boldsymbol{x}_t$ should be in the decreasing direction.

## 3.2 Pre-Training Generalized Diffusion Model

To create artificial functions that are similar to the target function, we will construct multiple Gaussian processes based on a diverse range of Gaussian process priors equipped with different (fixed) signal and length-scale parameters. Suppose we are given offline dataset $D = \{(\boldsymbol{x}_i, y_i)\}_{i=1}^n$ and kernel parameter $\boldsymbol{\phi}_s = \{\ell_s, \sigma_s\}$, the corresponding posterior is given in Eq. 4 above,

$$g_{\boldsymbol{\phi}_s}(\boldsymbol{x}) \sim \mathbb{N}\Big(\boldsymbol{k}(\boldsymbol{\phi}_s)^\top (\boldsymbol{K}(\boldsymbol{\phi}_s) + \sigma^2 \boldsymbol{I})^{-1} \boldsymbol{y}, \; k_{\boldsymbol{\phi}_s}(\boldsymbol{x}, \boldsymbol{x}) - \boldsymbol{k}(\boldsymbol{\phi}_s)^\top (\boldsymbol{K}(\boldsymbol{\phi}_s) + \sigma^2 \boldsymbol{I})^{-1} \boldsymbol{k}(\boldsymbol{\phi}_s)\Big) \tag{18}$$

For each such posterior, the mean function $\bar{g}_{\boldsymbol{\phi}_s}(\boldsymbol{x}) \triangleq \boldsymbol{k}(\boldsymbol{\phi}_s)^\top (\boldsymbol{K}(\boldsymbol{\phi}_s) + \sigma^2 \boldsymbol{I})^{-1} \boldsymbol{y}$ is similar to the target function at least at the offline data regime since the posterior is shaped based on the offline dataset. We will construct $\boldsymbol{X}_s^-$ and $\boldsymbol{X}_s^+$ as the set of low- and high-value inputs of $\bar{g}_{\boldsymbol{\phi}_s}(\boldsymbol{x})$ via running $M$ steps of gradient descent and ascent starting from the offline inputs, respectively:

$$\boldsymbol{X}_s^- \;\; \triangleq \;\; \left\{\boldsymbol{x}_M^- \triangleq \boldsymbol{x}_0^- - \eta \cdot \sum_{m=1}^{M} \nabla_{\boldsymbol{x}} \bar{g}_{\boldsymbol{\phi}_s}(\mathbf{x}_{m-1}^-) \;\; \Big| \;\; \boldsymbol{x}_0^- \in D\right\} \tag{19}$$

$$\boldsymbol{X}_s^+ \;\; \triangleq \;\; \left\{\boldsymbol{x}_M^+ \triangleq \boldsymbol{x}_0^+ + \eta \cdot \sum_{m=1}^{M} \nabla_{\boldsymbol{x}} \bar{g}_{\boldsymbol{\phi}_s}(\mathbf{x}_{m-1}^+) \;\; \Big| \;\; \boldsymbol{x}_0^+ \in D\right\}, \tag{20}$$

where $\boldsymbol{x}_{m+1}^+ \triangleq \boldsymbol{x}_m^+ + \eta \nabla_{\boldsymbol{x}} \bar{g}_{\boldsymbol{\phi}_s}(\boldsymbol{x}_m^+)$ and $\boldsymbol{x}_{m+1}^- \triangleq \boldsymbol{x}_m^- - \eta \nabla_{\boldsymbol{x}} \bar{g}_{\boldsymbol{\phi}_s}(\boldsymbol{x}_m^-)$. Here, $\boldsymbol{x}_0^+ = \boldsymbol{x}_0^- \in D$. We can then sample $(\boldsymbol{x}_T, \widehat{\boldsymbol{x}}_0) \sim \boldsymbol{X}_s^- \times \boldsymbol{X}_s^+$ as pre-training data to learn our diffusion process mapping between low- and high-value regions across the aforementioned posterior mean functions.

For practical implementation, we can also incorporate the corresponding outputs $\boldsymbol{y}_s^-$ and $\boldsymbol{y}_s^+$ of $\boldsymbol{X}_s^-$ and $\boldsymbol{X}_s^+$, respectively, as part of the input to the noise prediction network $\boldsymbol{\epsilon}_\theta(\boldsymbol{x}_t, t, y_s^+, y_s^-)$ in a similar manner to how we arrive at Eq. 11 from Eq. 10. Once this generalized diffusion model has been pre-trained using the synthetic data $\{(\boldsymbol{X}_s^+, \boldsymbol{y}_s^+), (\boldsymbol{X}_s^-, \boldsymbol{y}_s^-)\}_{s=1}^{n_g}$, where $n_g$ is the number of

Gaussian processes, we can use the transition kernel, $p_\theta(\boldsymbol{x}_{t-1} \mid \boldsymbol{x}_t, \boldsymbol{x}_T)$, of the backward diffusion process to map from the (implicit) offline data distribution (i.e., empirical distribution over the $Q = 128$ offline inputs with highest output) to a (better) distribution of high-performing inputs. The effectiveness of this approach is thoroughly evaluated in Section 4.

# 4 EXPERIMENTS

This section assesses the effectiveness of our proposed method, _Low- to High-Value Diffusion for Black-Box Optimization_, which is named **L2HD**. Extensive empirical evaluations have been conducted to rigorously compare the performance of **L2HD** against that of a diverse suite of recent baselines on a commonly used benchmark for offline optimization (Trabucco et al., 2022). First, our experiment setup is summarized in Section 4.1. Detailed results and observations are then discussed in Section 4.2. Furthermore, extensive ablation studies are presented in Section 4.3.

## 4.1 EXPERIMENTS SETTINGS

**Benchmark Tasks.** Our investigation covers four real-world tasks selected from the Design-Bench (Trabucco et al., 2022)[2]. The chosen tasks cover both discrete and continuous domains. The discrete tasks, **TF-Bind-8** and **TF-Bind-10** (Barrera et al., 2016b), aim to discover DNA sequences with high binding affinity to a specific transcription factor (SIX6 REF R1), with sequence lengths of 8 and 10, respectively. On the continuous side, **Ant Morphology** (Brockman et al., 2016) and **D'Kitty Morphology** (Ahn et al., 2020) focus on optimizing the physical structure of a simulated robot ant from OpenAI Gym (Brockman et al., 2016) and the D'Kitty robot from ROBEL (Ahn et al., 2020). For further details on these tasks, please read Appendix A.

**Baselines.** We selected a diverse group of 11 widely recognized offline optimizers for comparison, including **BO-qEI** (Trabucco et al., 2022), **CMA-ES** (Hansen), **REINFORCE** (Williams, 1992), **COMs** (Trabucco et al., 2021), **CbAS** (Brookes et al., 2019), **MINs** (Kumar & Levine, 2020a), **RoMA** (Yu et al., 2021), **DDOM** (Krishnamoorthy et al., 2023), **ICT** (Yuan et al., 2023), **Tri-mentoring**(Chen et al., 2024), and standard gradient ascent (**GA**).

**Evaluation Protocol.** Following the approach in (Trabucco et al., 2022), each method generates 128 optimized design candidates, which are then evaluated by the oracle function. The performances are ranked, and results are recorded at the 50[th], 80[th], and 100[th] percentiles. To ensure consistency, all results are averaged over 8 independent runs.

**Hyper-parameter Configuration.** For each baseline algorithm, we use the optimized hyper-parameters reported in the corresponding paper. In **L2HD**, there are several important hyper-parameters such as the number of gradient steps $M$, the step size $\eta$ during the data generation phase, as well as the conditional dropping probability $\kappa$ in Eq. 11. According to the result of fine-tuning, we achieve optimal performance with $M = 100$ and $\kappa = 0.15$ across all tasks. Additional ablation studies for the other hyperparameters of our framework are detailed in Section 4.3. For more details regarding the hyper-parameter configuration of **L2HD**, please read Appendix C.

## 4.2 EXPERIMENTAL RESULTS

In this section, we present a comparison of our method against 11 existing baselines. We have evaluated this at the 50-th, 80-th, and 100-th percentile levels. Due to space constraints, we only report results of the 100-th percentile level in the main text, while the results for the 50[th], 80[th] percentiles and score distribution of **L2HD** with others are provided in Appendix B.

**Results on Continuous Tasks:** The first two columns of Table 1 highlights **L2HD**'s performance on continuous tasks. Notably, we established a new state-of-the-art (SOTA) for the D'Kitty task, outperforming the runner-up baseline by over **0.022** in the mean score. For the Ant task, our model secured the runner-up position and surpassed the mean score of the third-ranked Tri-mentoring by a considerable margin **0.021**. In addition, the top-ranked CMA-ES exhibits a significantly larger

---

[2]We exclude tasks marked by previous works as having high oracle function noise and inaccuracy (**ChEMBL**, **Hopper**, and **Superconductor**) and tasks considered too expensive to evaluate (**NAS**)

Table 1: Experiments on Design-Bench Tasks. We report max score ($100^{th}$ percentile) among $Q = 128$ candidates. Blue denotes the best entry in the column, and Green denotes the second best. **Mean Rank** means the average rank of the method across all the benchmark datasets.

| | | Benchmarks | | | |
|---|---|---|---|---|---|
| **Method** | **Ant** | **D'Kitty** | **TFBind8** | **TFBind10** | **Mean Rank** |
| $\mathcal{D}_{offline}$ (best) | 0.565 | 0.884 | 0.439 | 0.467 | - |
| BO-qEI | 0.812 ± 0.000 | 0.896 ± 0.000 | 0.825 ± 0.091 | 0.627 ± 0.033 | 8.75 / 12 |
| CMA-ES | 1.561 ± 0.896 | 0.724 ± 0.001 | 0.939 ± 0.039 | 0.664 ± 0.034 | 4.75 / 12 |
| REINFORCE | 0.263 ± 0.026 | 0.573 ± 0.204 | 0.961 ± 0.034 | 0.618 ± 0.011 | 9.5 / 12 |
| GA | 0.293 ± 0.029 | 0.860 ± 0.021 | 0.985 ± 0.011 | 0.638 ± 0.032 | 6.75 / 12 |
| COMs | 0.882 ± 0.044 | 0.932 ± 0.006 | 0.940 ± 0.027 | 0.621 ± 0.033 | 6.5 / 12 |
| CbAS | 0.846 ± 0.033 | 0.895 ± 0.016 | 0.903 ± 0.028 | 0.649 ± 0.055 | 6.5 / 12 |
| MINs | 0.894 ± 0.022 | 0.939 ± 0.004 | 0.908 ± 0.063 | 0.630 ± 0.019 | 5.75 / 12 |
| RoMA | 0.593 ± 0.066 | 0.829 ± 0.020 | 0.665 ± 0.000 | 0.553 ± 0.000 | 11.0 / 12 |
| DDOM | 0.930 ± 0.029 | 0.925 ± 0.008 | 0.885 ± 0.061 | 0.634 ± 0.015 | 6.5 / 12 |
| ICT | 0.911 ± 0.030 | 0.945 ± 0.011 | 0.888 ± 0.047 | 0.624 ± 0.033 | 6.5 / 12 |
| Tri-mentoring | 0.944 ± 0.033 | 0.950 ± 0.015 | 0.899 ± 0.045 | 0.647 ± 0.039 | 4.25 / 12 |
| **L2HD (ours)** | 0.965 ± 0.014 | 0.972 ± 0.005 | 0.986 ± 0.007 | 0.685 ± 0.053 | 1.25 / 12 |

Table 2: Impact of gradient steps $M$ to the performance of **L2HD**.

| Steps ($M$) | Ant | DKitty | TFBind8 | TFBind10 |
|---|---|---|---|---|
| 25 | 0.968 ± 0.009 | 0.972 ± 0.003 | 0.952 ± 0.024 | 0.640 ± 0.039 |
| 50 | 0.968 ± 0.015 | 0.969 ± 0.003 | 0.945 ± 0.025 | 0.639 ± 0.024 |
| 75 | 0.950 ± 0.011 | 0.969 ± 0.005 | 0.972 ± 0.013 | 0.652 ± 0.033 |
| 100 | 0.965 ± 0.014 | 0.972 ± 0.005 | 0.986 ± 0.007 | 0.685 ± 0.053 |

Table 3: Impact of number of paired data points to the performance of **L2HD**.

| Initial Points ($n_p$) | Ant | TFBind8 |
|---|---|---|
| 128 | 0.915 ± 0.020 | 0.948 ± 0.024 |
| 256 | 0.950 ± 0.010 | 0.968 ± 0.018 |
| 512 | 0.964 ± 0.010 | 0.974 ± 0.006 |
| 1024 | 0.965 ± 0.014 | 0.986 ± 0.007 |

standard deviation of **0.896** compared to **L2HD** with **0.014** which is **64** times smaller. While CMA-ES performs well at the 100-th percentile, it receives much lower scores close to 0 at the 80-th and 50-th percentiles (as reported in Appendix B), highlighting the significant instability of this baseline. Moreover, our standard deviation for this task is lower than that of almost all other methods in the comparison, except for BO-qEI. This highlights the stability in the performance of **L2HD** .

**Results on Discrete Tasks:** The last two columns in Table 1 present the performance of **L2HD** on discrete tasks compared to other baseline methods. Our **L2HD** achieves both the top rank for the TFBind10 task with a mean score **0.685** and TFBind8 task with **0.986**. Our standard deviation **0.007** for the TFBind8 task is even smaller than that of all other baselines, except for RoMa, which performs poorly. These results demonstrate that **L2HD** remains a competitive and powerful method in discrete domains as well, confirming its effectiveness across both continuous and discrete tasks.

Overall, **L2HD** achieved an impressive mean rank of **1.25** across both discrete and continuous domains, setting a new SOTA on 3 out of 4 tasks. This demonstrates its consistent effectiveness.

### 4.3 ABLATION EXPERIMENTS

We additionally conducted ablation studies to investigate the impact of hyperparameters to the performance of our proposed method, **L2HD**.

**Number of Gradient Steps $M$.** We experimented with various numbers of gradient steps ($M$), from the set $\{25, 50, 75, 100\}$ to construct $\boldsymbol{X}_s^-$ and $\boldsymbol{X}_s^+$ during the data generation phase (see Section 3.2). Our experiments reveal that increasing $M$ consistently improves the overall performance of the algorithm as illustrated in Table 2. Increasing the number of gradient steps allows our model to more precisely distinguish between low-value and high-value regions in the distribution, substantially enhancing our performance. However, increasing the no. of gradient steps also increases computational time, so we select $M = 100$ as the best balance between performance and efficiency.

Table 4: Effectiveness of Gaussian process for generating synthetic data.

| | Type | Ant | DKitty | TFBind8 | TFBind10 |
|---|---|---|---|---|---|
| No-GP | 50th lowest - 50th highest percentile | 0.745 ± 0.097 | 0.952 ± 0.007 | 0.775 ± 0.057 | 0.641 ± 0.034 |
| | Lowest bins - Highest bins | 0.941 ± 0.020 | 0.952 ± 0.009 | 0.837 ± 0.055 | 0.651 ± 0.054 |
| GP | GP (1 function) | 0.955 ± 0.013 | 0.971 ± 0.004 | 0.984 ± 0.012 | 0.657 ± 0.029 |
| | GP (800 functions) | 0.965 ± 0.014 | 0.972 ± 0.005 | 0.986 ± 0.007 | 0.685 ± 0.053 |

Table 5: Performance of **L2HD** when changing the type of initial points.

| Type | Ant | TFBind8 |
|---|---|---|
| Random | 0.953 ± 0.014 | 0.976 ± 0.007 |
| Lowest | 0.545 ± 0.214 | 0.969 ± 0.009 |
| Highest | 0.965 ± 0.014 | 0.986 ± 0.007 |

Table 6: Experiments on Design-Bench Tasks in a few-shot experimental design setting. We report max score ($100^{\text{th}}$ percentile) among $Q = 128$ candidates.

| Method | Ant | TFBind8 |
|---|---|---|
| ExPT | 0.940 ± 0.027 | 0.874 ± 0.071 |
| **Ours** | 0.942 ± 0.035 | 0.895 ± 0.086 |

**Number of Initial Points.** For each synthetic function generated by the Gaussian process, we will draw a number of initial data points from the offline dataset to initiate the exploration into the low-value and high-value regions via gradient descent and ascent, respectively. We experimented with different numbers of initial points from the set $\{128, 256, 512, 1024\}$. As shown in the Table 3, this consistently led to improved performance. This observation is similar to a previous observation that having more well-curated training data tends to enhance the overall model performance. We selected 1024 as the best balance between computational cost and performance score.

**Effectiveness of Gaussian process for Generating Synthetic Data.** In addition to our main approach, we investigate alternative methods for generating synthetic data to train our generalized diffusion process. First, we explore two heuristic methods that do not depend on Gaussian processes (GP). The first method involves dividing the offline data into two bins: the 50th lowest percentile and the 50th highest percentile. From these bins, we sample pairs $(\boldsymbol{X}^-, \boldsymbol{X}^+)$, where $\boldsymbol{X}^-$ is sampled from the lowest percentile and $\boldsymbol{X}^+$ from the highest percentile. This method is referred to as 50th lowest - 50th highest percentile. The second method divides the data into 64 bins based on $\boldsymbol{y}$ values. In this case, $\boldsymbol{X}^-$ is sampled from the lowest bin, and $\boldsymbol{X}^+$ from the highest bin, referred to as Lowest bins - Highest bins. This approach is similar to the sampling strategy in (Krishnamoorthy et al., 2022), where trajectories with increasing outputs are selected from offline data to train a model that progressively guides designs from the lowest to the highest bin.

Additionally, we examine two methods that utilize GP for synthetic data generation. In the first scenario, synthetic training data is produced using only a single function derived from a Gaussian process, referred to as GP(1 function). The second scenario corresponds to our proposed method, where multiple functions derived from various Gaussian processes are used to generate synthetic data. The results presented in Table 4 demonstrate that generating synthetic data using Gaussian processes consistently outperforms heuristic methods. Moreover, utilizing multiple functions from various Gaussian processes achieves the best performance.

**Changing the Type of Initial Points.** We explore three strategies for selecting initial points during the data generation phase. The first strategy randomly samples points from the offline data. The second selects points with the lowest objective values, while the third chooses points with the highest objective values. For all strategies, we perform M steps of gradient ascent and descent on these initial points to generate low-value and high-value data, respectively. During the adaptation phase, we select the top 128 points with the highest values from the offline data to represent the (implicit) distribution of low-value designs. Sequential denoising steps are then applied to approach a better distribution in the higher-output region of the oracle data. As shown in Table 5, the third strategy delivers the best performance.

**Few-Shot Experimental Designs Setting.** In offline optimization, the few-shot experimental designs (ED) setting was introduced in ExPT (Nguyen et al., 2023), presenting a more challenging task scenario. It is assumed that only a few labeled data points $D_{\text{label}} = \{(\boldsymbol{x}_i, y_i)\}_{i=1}^{n}$ are accessible alongside a larger set of unlabelled data points $D_{\text{unlabeled}} = \{\tilde{\boldsymbol{x}}_i\}_{i=1}^{m}$ for offline optimization. To evaluate our model in this setting, we follow ExPT's protocol which utilizes a random 1% of

the offline data as labeled data points and considers the remaining $99\%$ as unlabeled. To adapt the generation of synthetic functions for this scenario, we first fit a Gaussian Process (GP) to $D_{\text{label}}$, generate pseudo-labels for $D_{\text{unlabeled}}$, and then refit the GP to the combined dataset. The mean function of this refitted GP is used as the synthetic function, following the same steps as the main method. As demonstrated in Table 6, our method also outperformed ExPT model in this setting.

## 5 RELATED WORKS

Existing approaches in offline optimization can be categorized into three main families: **forward modeling**, **inverse modeling**, and **learning search policies**.

**Forward Modeling** tackles out-of-distribution (OOD) issues by penalizing high surrogate predictions on OOD inputs (Trabucco et al., 2021; Chen et al., 2024; Yuan et al., 2023; Yu et al., 2021; Dao et al., 2024; Hoang et al., 2024; Qi et al., 2022; Fu & Levine, 2021; Fannjiang & Listgarten, 2020). For example, (Trabucco et al., 2021) identifies OOD regions early during gradient updates and retrains the surrogate with regularizers to penalize high-value predictions at these inputs. Dao et al. (2024) introduces a sensitivity-aware regularizer for offline optimizers, while Yuan et al. (2023); Chen et al. (2024) use co-teaching among surrogates to improve performance.

**Inverse Modeling** avoids OOD problems by directly learning high-value regions (Kumar & Levine, 2020a; Nguyen et al., 2023; Krishnamoorthy et al., 2023). For instance, Kumar & Levine (2020a) uses model inversion networks (MINs) to map scores back to inputs, while Nguyen et al. (2023) combines unsupervised learning and few-shot experimental design for optimizing synthetic functions. Krishnamoorthy et al. (2023) develops a guided diffusion model to generate designs conditioned on function values. The model is trained using weighted sampling from the offline dataset.

**Learning Search Policies** aims to replicate optimization paths from low- to high-value designs (Krishnamoorthy et al., 2022; Chemingui et al., 2024). Krishnamoorthy et al. (2022) synthesizes trajectories from offline data using a heuristic for monotonic transitions and trains an auto-regressive model. Chemingui et al. (2024) reinterprets offline optimization as a reinforcement learning task, which optimizes for an effective policy using sampled trajectories from the offline dataset.

Overall, these methods remain limited by the available amount of offline data. For example, Krishnamoorthy et al. (2023) uses guided diffusion to learn an inverse mapping from a desired performance output to potential input designs. The adopted diffusion model, however, has to be trained on weighted sampling from the offline data, which might not contain important information regarding potential high-performing input regions far from the offline regimes.

To mitigate this data bottleneck, we instead investigate a revisitation of offline optimization as a distributional translation task. This interestingly shows the equivalence between a direct distributional augmentation of the surrogate-based gradient update (to compensate for the distributional gap between the offline and oracle data) and a generalized diffusion model that, unlike previous diffusion models, can map between two (implicit) data distributions. More interestingly, our proposed approach can leverage external guiding information from related functions to learn the diffusion model without requiring access to samples from the target distribution. This is essential in our context since the offline dataset is often sampled from a distribution of low-value inputs.

## 6 CONCLUSION

To conclude, we have reformulated offline optimization as a distributional translation task. Our key finding here is the correspondence between a direct stochastic augmentation of a conventional surrogate-based gradient ascent and an equivalent perspective on generalized diffusion. This results in a principled model to map from the offline input distribution to another distribution over high-performing input regions. We adopt a robust learning framework of pre-training and adaptation for the derived diffusion model in which additional guiding information can be extracted from related domains to improve its training efficacy. This further reveal a potential future direction to build pre-trained foundation model for offline optimization which can be fast adapted to optimize effectively any emerging tasks with sparse data. Our empirical results demonstrate the effectiveness of this method, setting a new state-of-the-art performance for offline optimization.

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

Table 7: Overview of tasks in the Design-Bench benchmark.

| Task | Offline Size | Dimensions | Categories | Type | Oracle |
|------|-------------|------------|------------|------|--------|
| TF Bind 8 | 32,898 | 8 | 4 | Discrete | Exact |
| TF Bind 10 | 10000 | 10 | 4 | Discrete | Exact |
| Ant Morphology | 10004 | 60 | N/A | Continuous | Exact |
| D'Kitty Morphology | 10004 | 56 | N/A | Continuous | Exact |

## A  TASK DETAILS

Design-Bench (Trabucco et al., 2022) is a widely adopted benchmark for evaluating offline black-box optimization algorithms. Table 7 presents the summary of four benchmark tasks in Design-Bench.

**TF Bind 8 and TF Bind 10: DNA sequence optimization.** The aim of TF Bind 8 and TF Bind 10 benchmarks is to discover the length-8 DNA sequence with the strongest binding affinity to a specific transcription factor(SIX6_REF_R1 by default) (Barrera et al., 2016a). In the TF Bind8 benchmark, the binding affinities cover all 65,792 possible sequences. Those are 1,048576 sequences for the TF Bind 10 benchmark. A sequence is made up of four possible nucleotides, each representing a categorical choice. In the TF Bind 8, an offline dataset of 32,898 sequences is sampled, while in the TF Bind 10, we use a set of 10,000 sequences for forming offline data.

**Ant and D'Kitty Morphology: robot morphology optimization.** This task focuses on optimizing the physical structure of two simulated robots: Ant from OpenAI Gym Brockman et al. (2016) and D'Kitty from ROBEL Ahn et al. (2019). For Ant Morphology, the objective is to enhance the structure of a quadruped robot to maximize its running speed. In D'Kitty Morphology the goal is to improve the D'Kitty robot's structure to enable it to reach a specific target. Both tasks aim to discover optimal robot morphologies for their respective challenges. To control the robots with the newly optimized structures, a controller is trained using the Soft Actor-Critic algorithm Haarnoja et al. (2018), tailored for each morphology. The morphology parameters, such as the size, orientation, and placement of limbs, result in 60 continuous variables for Ant and 56 for D'Kitty. The evaluation process runs simulations in MuJoCo Todorov et al. (2012) for 100 time steps, averaging results from 16 independent trials to obtain accurate yet computationally efficient performance estimates.

## B  ADDITIONAL EXPERIMENT RESULTS

In the main manuscript, we reported the 100th percentile scores. Here we present scores at $80^{th}$ and $50^{th}$ percentiles, providing additional insights into the performance distribution of our **L2HD** across different levels of evaluation.

### B.1  PERFORMANCE EVALUATION AT $80^{TH}$ PERCENTILE LEVEL OF **L2HD**

As shown in Table 8, our **L2HD** consistently demonstrates impressive performance at the $80^{th}$ percentile level, achieving the best mean rank, i.e., **1.5**. Notably, in the Ant and Dkitty tasks, we exhibit significant improvements over all other baselines, with very small standard deviations of **0.005** and **0.002** respectively. Our **L2HD** model stands out in these tasks, with score differences of **0.098** and **0.030** compared to the runner-ups for Ant and Dkitty tasks respectively. For TFBind8, we continue to secure the best rank, outperforming the GA with a margin of **0.011**. It strongly demonstrates the reliability and stability of our model's performance.

### B.2  PERFORMANCE EVALUATION AT $50^{TH}$ PERCENTILE LEVEL OF **L2HD**

According to Table 9, our **L2HD** model achieves the best mean rank of **2.0** at the $50^{th}$ percentile level. At this score level, we surpass the Gradient Ascent (GA) in the TFBind8 task with a significant score difference of **0.072**, securing the top rank among all other methods. In the Ant and Dkitty tasks, we continue to dominate the other baselines with score differences of **0.094** and **0.030** compared to the runner-ups, respectively. Furthermore, we achieve standard deviations **0.015** and **0.002** which is

Table 8: Experiments on Design-Bench Tasks. We report $80^{\text{th}}$ percentile score among $Q = 128$ candidates. Blue denotes the best entry in the column, and Green denotes the second best. **Mean Rank** means the average rank of the method over all the experiment benchmarks.

| Method | Benchmarks | | | | Mean Rank |
|---|---|---|---|---|---|
| | **Ant** | **D'Kitty** | **TFBind8** | **TFBind10** | |
| $\mathcal{D}_{offline}$ (best) | 0.565 | 0.884 | 0.439 | 0.467 | - |
| BO-qEI | 0.629 ± 0.000 | 0.884 ± 0.000 | 0.439 ± 0.000 | 0.510 ± 0.011 | 9.0 / 12 |
| CMA-ES | 0.007 ± 0.013 | 0.718 ± 0.001 | 0.652 ± 0.017 | 0.543 ± 0.013 | 7.75 / 12 |
| REINFORCE | 0.182 ± 0.017 | 0.562 ± 0.197 | 0.622 ± 0.030 | 0.519 ± 0.007 | 9.5 / 12 |
| GA | 0.189 ± 0.014 | 0.762 ± 0.036 | 0.828 ± 0.027 | 0.516 ± 0.004 | 7.75 / 12 |
| COMs | 0.635 ± 0.031 | 0.887 ± 0.004 | 0.738 ± 0.027 | 0.526 ± 0.012 | 4.25 / 12 |
| CbAS | 0.542 ± 0.034 | 0.813 ± 0.012 | 0.585 ± 0.030 | 0.517 ± 0.008 | 8.5 / 12 |
| MINs | 0.746 ± 0.011 | 0.908 ± 0.004 | 0.545 ± 0.031 | 0.519 ± 0.010 | 5.5 / 12 |
| RoMA | 0.298 ± 0.033 | 0.738 ± 0.018 | 0.661 ± 0.010 | 0.525 ± 0.003 | 7.5 / 12 |
| DDOM | 0.749 ± 0.029 | 0.865 ± 0.009 | 0.526 ± 0.017 | 0.506 ± 0.004 | 8.0 / 12 |
| ICT | 0.708 ± 0.019 | 0.898 ± 0.004 | 0.667 ± 0.035 | 0.525 ± 0.016 | 4.75 / 12 |
| Tri-mentoring | 0.722 ± 0.015 | 0.902 ± 0.003 | 0.683 ± 0.047 | 0.531 ± 0.007 | 3.25 / 12 |
| **Ours** | 0.847 ± 0.005 | 0.938 ± 0.002 | 0.839 ± 0.015 | 0.526 ± 0.007 | 1.5 / 12 |

much lower than other competitive methods, reinforcing the power and consistency of our method across these tasks.

### B.3 Score distribution of L2HD

We combine the candidates from 8 runs ($128 \times 8 = 1024$ designs) to visualize the distribution of scores for **L2HD** in comparison to other baselines. The results are shown in Figure 2. Notably, we split the Ant task into two figures due to the wide range of CMA-ES results, and in each figure, we plot the **L2HD** max and **L2HD** median lines for comparison with the baselines. Figure 2 reveals that **L2HD** has a score distribution skewed towards higher-value regions, particularly in the D'Kitty task. In the Ant task, although the **L2HD** max score is not as high as that of CMA-ES, CMA-ES achieves a single best design while the others perform poorly, as also observed in Table 8 and Table 9. Additionally, **L2HD** demonstrates a superior score distribution compared to other methods in two discrete tasks: TFBind8 and TFBind10.

**Summary**

In conclusion, the reported results demonstrate that our **L2HD** model maintains its position as the best-performing model across levels of reported scores. This consistent performance not only highlights the effectiveness and stability of **L2HD** but also reaffirms the reliability of our novel approach to the model-based optimization problem.

## C  DETAIL ALGORITHM AND IMPLEMENTATION OF **L2HD** MODEL

### C.1  GENERATING SYNTHETIC DATA WITH GP

For generating our synthetic function, we first sample the parameters $l$ (lengthscale) and $\sigma^2$ (variance) uniformly from the ranges $[l_0 - \delta, l_0 + \delta]$ and $]\sigma_0^2 - \delta, \sigma_0^2 + \delta]$, where $l_0$, $\sigma_0^2$ and $\delta$ are fixed initial hyperparameters, as reported in Table 10. After sampling, we compute the mean function of the Gaussian Process posterior. We then sample $n_p$ points from the offline data and perform $M = 100$ gradient ascent and gradient descent steps with a step size $\eta$ (more details are presented in Section 3.2). To enhance the quality of our synthetic data, we exclude any pair of points $(\boldsymbol{x}^-, y^-)$ and $(\boldsymbol{x}^+, y^+)$ with a small objective value margin by a filter. Specifically, if the difference between $y^+$ and $y^-$ is smaller than a threshold, those pairs will be filtered out from the synthetic data. All key hyperparameters for this process are listed in Table 10. The detailed algorithm is represented in Algorithm 2.

Table 9: Experiments on Design-Bench Tasks. We report $50^{\text{th}}$ percentile score among $Q = 128$ candidates. Blue denotes the best entry in the column, and Green denotes the second best. **Mean Rank** means the average rank of the method over all the experiment benchmarks.

| | Benchmarks | | | | Mean Rank |
|---|---|---|---|---|---|
| **Method** | **Ant** | **D'Kitty** | **TFBind8** | **TFBind10** | |
| $\mathcal{D}_{offline}$ (best) | 0.565 | 0.884 | 0.439 | 0.467 | - |
| BO-qEI | 0.569 ± 0.000 | 0.883 ± 0.000 | 0.439 ± 0.000 | 0.469 ± 0.005 | 6.25 / 12 |
| CMA-ES | -0.043 ± 0.007 | 0.674 ± 0.016 | 0.536 ± 0.012 | 0.490 ± 0.015 | 7.25 / 12 |
| REINFORCE | 0.140 ± 0.026 | 0.510 ± 0.203 | 0.450 ± 0.024 | 0.470 ± 0.010 | 9.0 / 12 |
| GA | 0.137 ± 0.014 | 0.591 ± 0.132 | 0.603 ± 0.045 | 0.469 ± 0.006 | 7.5 / 12 |
| COMs | 0.471 ± 0.034 | 0.862 ± 0.003 | 0.598 ± 0.031 | 0.475 ± 0.010 | 4.75 / 12 |
| CbAS | 0.369 ± 0.008 | 0.748 ± 0.016 | 0.441 ± 0.021 | 0.465 ± 0.006 | 8.75 / 12 |
| MINs | 0.618 ± 0.016 | 0.889 ± 0.003 | 0.421 ± 0.017 | 0.467 ± 0.010 | 6.0 / 12 |
| RoMA | 0.224 ± 0.020 | 0.545 ± 0.170 | 0.519 ± 0.073 | 0.518 ± 0.003 | 7.0 / 12 |
| DDOM | 0.568 ± 0.066 | 0.814 ± 0.016 | 0.404 ± 0.012 | 0.456 ± 0.002 | 9.0 / 12 |
| ICT | 0.554 ± 0.018 | 0.872 ± 0.007 | 0.557 ± 0.031 | 0.457 ± 0.033 | 6.75 / 12 |
| Tri-mentoring | 0.572 ± 0.016 | 0.884 ± 0.001 | 0.562 ± 0.051 | 0.475 ± 0.009 | 3.25 / 12 |
| **Ours** | 0.712 ± 0.014 | 0.919 ± 0.003 | 0.675 ± 0.026 | 0.473 ± 0.004 | 2.0 / 12 |

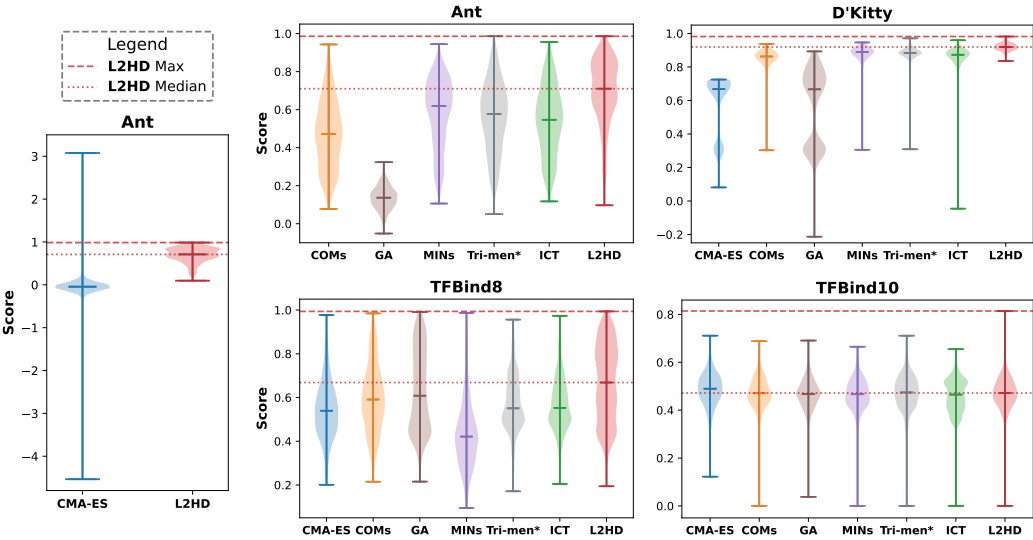

Figure 2: Score distribution of found candidates of **L2HD** compared to others. (**Tri-men\*** stands for **Tri-mentoring**).

## C.2 TRAINING THE BROWNIAN BRIDGE DIFFUSION MODEL

In this section, we describe how BBDM Li et al. (2023) is employed to handle our distributional translation. From Eq. 15, setting $\zeta_t = 1 - m_t = 1 - t/T$ and $\delta_t = 2(m_t - m_t^2)$ leads to a variance-preservation version of Brownian bridge process introduced in the paper BBDM Li et al. (2023):

$$\boldsymbol{x}_t = (1 - m_t)\widehat{\boldsymbol{x}}_0 + m_t \boldsymbol{x}_T + \sqrt{\delta_t}\epsilon_t \tag{21}$$

The above equation serves as a parameterized form of $q(\boldsymbol{x}_t | \widehat{\boldsymbol{x}}_0, \boldsymbol{x}_T)$. Now, using this form at timestep $t - 1$,

$$\boldsymbol{x}_{t-1} = (1 - m_{t-1})\widehat{\boldsymbol{x}}_0 + m_{t-1}\boldsymbol{x}_T + \sqrt{\delta_{t-1}}\epsilon_{t-1} \tag{22}$$

---

**Algorithm 1** : Low- to High-Value Diffusion for Offline Optimization ( **L2HD** )

---

**Input:** Offline dataset $D = \{\boldsymbol{x}_i, y_i\}_{i=1}^N$; no. of epochs $E$; no. of diffusion steps $T$, scale number $\alpha$, best objective value $y_*$, conditional dropout probability $\kappa$, no. of iterations $I$.
**Output:** High-value candidate $\widehat{\boldsymbol{x}}_0$.
 1: **Phase 1:** Pretraining
 2: Initialize model parameters $\theta^0$
 3: **for** $e = 1 \rightarrow E$ **do**
 4:     Generate synthetic dataset $D_{syn}$ from Algorithm 2
 5:     **for** $i = 1 \rightarrow I$ **do**
 6:         Sample $\{\widehat{\boldsymbol{x}}_0, y_0, \boldsymbol{x}_T, y_T\} \sim \mathcal{D}_{syn} = \{\boldsymbol{X}^+, \boldsymbol{y}^+, \boldsymbol{X}^-, \boldsymbol{y}^-\}$
 7:         Sample timestep $t \sim Uniform(1, T)$, $\gamma \sim Ber(\kappa)$, $y = [y_0, y_T]$.
 8:         Forward diffusion $\boldsymbol{x}_t = (1 - m_t) \cdot \widehat{\boldsymbol{x}}_0 + m_t \cdot \boldsymbol{x}_T + \sqrt{\delta_t}\epsilon$ where $\epsilon \sim \mathbb{N}(0, \boldsymbol{I})$
 9:         Take gradient descend step on
            $\nabla_\theta \| m_t(\boldsymbol{x}_T - \widehat{\boldsymbol{x}}_0) + \sqrt{\delta_t}\boldsymbol{\epsilon} - \boldsymbol{\epsilon}_\theta(\boldsymbol{x}_t, t, (1 - \gamma) \cdot y + \gamma \cdot \emptyset) \|^2$ – Eq. 30
10:     **end for**
11: **end for**
12:
13: **Phase 2:** Zero-shot Adaptation
14: $\{\boldsymbol{x}_T, y_T\} \leftarrow$ 128 best designs in $D$, $y = [\alpha \cdot y_*, y_T]$
15: **for** $t = T \rightarrow 1$ **do**
16:     $z \sim \mathbb{N}(0, \boldsymbol{I})$ if $t > 0$ else $z = 0$
17:     Compute $\epsilon_\theta(\boldsymbol{x}_t, t, y)$ via Eq. 31
18:     $\boldsymbol{x}_{t-1} = i_t \cdot \boldsymbol{x}_t + j_t \cdot \boldsymbol{x}_T + k_t \cdot \epsilon_\theta(\boldsymbol{x}_t, t, y) + \sqrt{\delta_t} \cdot z$
19: **end for**
20: **return** high-value design $\widehat{\boldsymbol{x}}_0 = \boldsymbol{x}_0$.

---

**Algorithm 2** : Synthetic Data Generation with Gaussian process

---

**Input:** Offline dataset $D = \{\boldsymbol{x}_i, y_i\}_{i=1}^N$; no. of function per epoch $n_f$; no. of points $n_p$; no. of gradient steps $M$
**Output:** Synthetic data $D_{syn} = \{\boldsymbol{X}^-, \boldsymbol{y}^-, \boldsymbol{X}^+, \boldsymbol{y}^+\}$
 1: $D_{syn} = \emptyset$
 2: **for** $s = 1 \rightarrow n_f$ **do**
 3:     Sample uniformly $l_s \sim U(l_0 - \delta, l_0 + \delta)$, $\sigma_s^2 \sim U(\sigma_0^2 - \delta, \sigma_0^2 + \delta)$
 4:     Compute the mean function $\bar{g}_{\phi_s}$ of posterior Gaussian process via Eq. 18
 5:     Sample $n_p$ top points from offline data $\{\boldsymbol{x}_0^i, y_0^i\}_{i=1}^{n_p} \sim D$
 6:     Compute low-value design $\boldsymbol{X}_s^-$ via Eq. 19
 7:     Compute high-value design $\boldsymbol{X}_s^+$ via Eq. 20
 8:     Compute corresponding low and high scores $\boldsymbol{y}_s^-, \boldsymbol{y}_s^+ = \bar{g}_{\phi_s}(\boldsymbol{X}_s^-), \bar{g}_{\phi_s}(\boldsymbol{X}_s^+)$
 9:     $D_{syn} = D_{syn} \bigcup \{\boldsymbol{X}_s^-, \boldsymbol{y}_s^-, \boldsymbol{X}_s^+, \boldsymbol{y}_s^+\}$
10: **end for**
11: **return** synthetic data $D_{syn}$

---

Substituting expression of $\widehat{\boldsymbol{x}}_0$ from Eq. 22 to Eq. 21, leading to

$$q(\boldsymbol{x}_t | \boldsymbol{x}_{t-1}, \boldsymbol{x}_T) = \mathbb{N}\left( \frac{1 - m_t}{1 - m_{t-1}} \boldsymbol{x}_{t-1} + \left( m_t - \frac{1 - m_t}{1 - m_{t-1}} \right) \boldsymbol{x}_T, \delta_{t|t-1} \mathbb{I} \right) \tag{23}$$

where $\delta_{t|t-1}$ is computed as

$$\delta_{t|t-1} = \delta_t - \delta_{t-1} \left( \frac{1 - m_t}{1 - m_{t-1}} \right)^2 \tag{24}$$

By combining Eq. 23 and Eq. 21, we can derive $q(\boldsymbol{x}_{t-1} | \boldsymbol{x}_t, \widehat{\boldsymbol{x}}_0, \boldsymbol{x}_T)$ through Bayes' theorem and Markov chain property:

$$
\begin{aligned}
q(\boldsymbol{x}_{t-1} | \boldsymbol{x}_t, \widehat{\boldsymbol{x}}_0, \boldsymbol{x}_T) &= \frac{q(\boldsymbol{x}_t | \boldsymbol{x}_{t-1}, \boldsymbol{x}_T) \cdot q(\boldsymbol{x}_{t-1} | \widehat{\boldsymbol{x}}_0, \boldsymbol{x}_T)}{q(\boldsymbol{x}_t | \widehat{\boldsymbol{x}}_0, \boldsymbol{x}_T)} \\
&= \mathbb{N}(\boldsymbol{x}_{t-1}; \tilde{\boldsymbol{\mu}}(\boldsymbol{x}_t, \widehat{\boldsymbol{x}}_0, \boldsymbol{x}_T), \tilde{\delta}_t \cdot \boldsymbol{I})
\end{aligned}
\tag{25}
$$

Table 10: Hyperparameters for the synthetic data generation of **L2HD**.

| Hyperparameter | Value |
|---|---|
| $l_0, \sigma_0^2$ | 1.0 (continuous) 
 6.25 (discrete) |
| $\delta$ | 0.25 |
| Step size ($\eta$) | 0.001 (continuous) 
 0.05 (discrete) |
| Number of gradient steps ($M$) | 100 |
| Threshold | 0.001 |

where the variance $\tilde{\delta}_t = \left( \delta_t - \delta_{t-1} \frac{(1-m_t)^2}{(1-m_{t-1})^2} \right) \frac{\delta_{t-1}}{\delta_t}$ and the mean

$$\tilde{\boldsymbol{\mu}}(\boldsymbol{x}_t, \widehat{\boldsymbol{x}}_0, \boldsymbol{x}_T) \quad = \quad i_t \cdot \boldsymbol{x}_t + j_t \cdot \boldsymbol{x}_T + k_t \cdot \left( m_t(\boldsymbol{x}_T - \widehat{\boldsymbol{x}}_0) + \sqrt{\delta_t}\boldsymbol{\epsilon} \right) \tag{26}$$

with

$$i_t \quad = \quad \frac{\delta_{t-1}}{\delta_t} \frac{1-m_t}{1-m_{t-1}} + \frac{\delta_t - \delta_{t-1} \frac{(1-m_t)^2}{(1-m_{t-1})^2}}{\delta_t}(1-m_{t-1})$$

$$j_t \quad = \quad m_{t-1} - m_t \frac{1-m_t}{1-m_{t-1}} \frac{\delta_{t-1}}{\delta_t}$$

$$k_t \quad = \quad (1-m_{t-1}) \frac{\delta_t - \delta_{t-1} \frac{(1-m_t)^2}{(1-m_{t-1})^2}}{\delta_t}$$

Regarding reverse process, BBDM parameterizes the $p_\theta(\boldsymbol{x}_{t-1}|\boldsymbol{x}_t, \boldsymbol{x}_T)$ as:

$$p_\theta(\boldsymbol{x}_{t-1}|\boldsymbol{x}_t, \boldsymbol{x}_T) = \mathbb{N}(\boldsymbol{x}_{t-1}; \boldsymbol{\mu}_\theta(\boldsymbol{x}_t, \boldsymbol{x}_T, t), \tilde{\delta}_t \cdot \boldsymbol{I}) \tag{27}$$

where the $\mu_\theta$ is parameterized following the Eq. 26:

$$\boldsymbol{\mu}_\theta(\boldsymbol{x}_t, \boldsymbol{x}_T, t) = i_t \cdot \boldsymbol{x}_t + j_t \cdot \boldsymbol{x}_T + k_t \cdot \boldsymbol{\epsilon}_\theta(\boldsymbol{x}_t, t) \tag{28}$$

Therefore, the training objective ELBO in Eq. 17 can be simplified as:

$$\mathbb{E}_{\widehat{\boldsymbol{x}}_0, \boldsymbol{x}_T, \boldsymbol{\epsilon}} \left[ \| m_t(\boldsymbol{x}_T - \widehat{\boldsymbol{x}}_0) + \sqrt{\delta_t}\boldsymbol{\epsilon} - \boldsymbol{\epsilon}_\theta(\boldsymbol{x}_t, t) \|^2 \right] \tag{29}$$

For practical implementation, we leverage the synthetic data for our BBDM training, particularly $\{\widehat{\boldsymbol{x}}_0, y_0, \boldsymbol{x}_T, y_T\} \sim \mathcal{D}_{syn} = \{\boldsymbol{X}^+, \boldsymbol{y}^+, \boldsymbol{X}^-, \boldsymbol{y}^-\}$, where $y_0, y_T$ is corresponding score of $\widehat{\boldsymbol{x}}_0, \boldsymbol{x}_T$. In addition, we integrate $y = (y_0, y_T)$ as a condition with the classifier-free guidance diffusion technique that is similar to Eq. 11:

$$\theta^* \triangleq \arg\min_\theta \mathbb{E}_{\widehat{\boldsymbol{x}}_0, y_0, \boldsymbol{x}_T, y_T, \boldsymbol{\epsilon}} \left[ \| m_t(\boldsymbol{x}_T - \widehat{\boldsymbol{x}}_0) + \sqrt{\delta_t}\boldsymbol{\epsilon} - \boldsymbol{\epsilon}_\theta(\boldsymbol{x}_t, t, (1-\gamma) \cdot y + \gamma \cdot \emptyset) \|^2 \right] \tag{30}$$

where $\gamma \sim Ber(\kappa)$. After training, the noise network $\epsilon_\theta$ is computed as below in the zero-shot adaptation phase:

$$\epsilon_\theta(\boldsymbol{x}_t, t, y) = (1+\beta) \cdot \epsilon_\theta(\boldsymbol{x}_t, t, y) - \beta \cdot \epsilon_\theta(\boldsymbol{x}_t, t) \tag{31}$$

where $\beta$ is the classifier free guidance weight and $y = [\alpha \cdot y_*, y_T]$ with $y_*$ is the maximum oracle score to generate high-value design $\widehat{\boldsymbol{x}}_0$. Our detail algorithm is represented in Algorithm 1.

For hyperparameters, we utilize an MLP $\boldsymbol{\epsilon}_\theta(\boldsymbol{x}_t, t, y)$ comprising four layers, each with a hidden size of 1024. Each layer employs the Swish activation function, defined as $\text{Swish}(z) = z\sigma(z)$, where $\sigma(z)$ is the sigmoid function. The MLP is trained using the Adam optimizer for 100 epochs with a learning rate of 0.001. During each epoch, we sample $n_f = 8$ synthetic functions from the Gaussian process and generate $n_p = 1024$ samples for each function. At the testing phase, we sample high-design candidates from the 128 best designs in the offline data, using $T = 200$ sequential denoising steps. All hyperparameters for modeling, training, and sampling with our model are summarized in Table 11.

For more details on the implementation, you can also check out our code provided at the following link: https://anonymous.4open.science/r/ICLR25-A100-03EA

Table 11: Hyperparameters for the Generalized diffusion process in **L2HD** .

| | **Hyperparameter** | **Value** |
|---|---|---|
| Architecture | Hidden size | 1024 |
| | Number of layers | 4 |
| | Activation | Swish |
| Training | Number of epochs | 100 |
| | Number of functions (per one epoch) | 8 |
| | Number of data points (per one function) | 1024 |
| | Learning rate | 0.001 |
| | Optimizer | Adam |
| | Batch size | 64 |
| | Conditional dropout ($\kappa$) | 0.15 |
| Sampling , | $\alpha$ | 0.8 |
| | $\beta$ | -1.5 |
| | Denoising steps | 200 |