# OpenReview forum: "FROM LOW TO HIGH-VALUE DESIGNS: OFFLINE OPTIMIZATION VIA GENERALIZED DIFFUSION"
_ICLR.cc/2025/Conference — Submitted to ICLR 2025_

### Official Review · Reviewer_3WMn · 2024-11-02

**Soundness:** 3
**Presentation:** 3
**Contribution:** 3
**Rating:** 6
**Confidence:** 3

**Summary:**

This paper presents a new perspective on offline optimization, framing the optimization task as a distributional translation from low-value points to high-value points. It uses diffusion models to learn this translation. The paper shows the effectiveness of this new perspective through theoretical analysis and experimental validation.

**Strengths:**

1. The new perspective of viewing the optimization problem as a diffusion translation problem is quite novel and intuitive.

2. The method proposed in this paper is direct and clear: It uses synthetic data from fitted gaussian process for data augmentation and then employs a diffusion model to learn the distributional translation from low-value to high-value regions.

3. The presentation of this paper is clear and well-structured.

4. The experiments are comprehensive.

**Weaknesses:**

The method needs to fit multiple Gaussian processes to generate synthetic data, which might have high time complexity in high-dimensional cases. Besides, the reverse diffusion process also requires multiple denoising steps, which also increases the time complexity.

**Questions:**

NA

---

> ### Author Response · Authors · 2024-11-22
> **Response to Reviewer 3WMn**
>
> Thank you very much for your positive review and acceptance rating. We address your question regarding the time-complexity of our method in high-dimensional cases below:
>
> >The method needs to fit multiple Gaussian processes to generate synthetic data, which might have high time complexity in high-dimensional cases.
>
> We note that fitting a GP requires (1) evaluating Eq. (5) and (2) differentiating the RHS of Eq. (5) with respect to each kernel parameter. Given n training data points in a $d$-dimensional space, the computational cost of (1) is $O(n^3 + dn^2)$ while that of (2) is $O(|$#kernel-parameter$|n^3) = O(n^3)$ since in our case, #kernel-parameter = 2 -- see lines 138-139.
>
> Hence, the cost of fitting a GP is linear in $d$ which will allow our proposed method (L2HD) to scale well to high-dimensional cases. To see this, we have detailed below the running time of our algorithm and other baselines across our experimented datasets.
>
> | Model         | Ant   | D'Kitty | TFBind8 | TFBind10 |
> |---------------|-------|--------|---------|----------|
> | GA            | 122s  | 200s   | 124s    | 420s     |
> | BO-QEI        | 219s  | 310s   | 402s    | 383s     |
> | CMA-ES        | 5747s | 4889s  | 2667s   | 4323s    |
> | MINS          | 797s  | 998s   | 2213s   | 792s     |
> | REINFORCE     | 240s  | 257s   | 507s    | 373s     |
> | CBAS          | 394s  | 390s   | 835s    | 529s     |
> | ICT           | 188s  | 189s   | 216s    | 639s     |
> | Tri-mentoring | 4276s | 3921s  | 4673s   | 3410s    |
> | **L2HD**          | 298s  | 297s   | 407s    | 575s     |
>
> The reported running time shows that even with the GP fitting overhead, L2HD still runs faster than several baselines such as CMA-ES, MINS, and Tri-mentoring.
>
> Furthermore, to avoid dealing with high-dimensional data, we can also adopt a similar approach used in BBDM which uses VQGAN to embed data into a lower-dimensional latent space first.
>
> >Besides, the reverse diffusion process also requires multiple denoising steps, which also increases the time complexity.
>
> We agree that the reverse diffusion process might require thousands of denoising steps, increasing inference time complexity. But again, the complexity only scales linearly in the number of denoising steps, and similar to BBDM, we find that using 200 denoising steps is sufficient to guarantee good performance across all tasks. The corresponding processing time -- as reported in the above table -- therefore remains within range in comparison to those of the baselines.
>
> We hope that our answer has addressed your concern. If so, we hope you would consider increasing the overall rating of our work. Otherwise, please let us know if you have any follow-up questions for us.

---

> > ### Comment · Reviewer_3WMn · 2024-11-25
> >
> > Thank you for your response. I'll keep my score.

---

### Official Review · Reviewer_7UEw · 2024-11-03

**Soundness:** 2
**Presentation:** 2
**Contribution:** 2
**Rating:** 5
**Confidence:** 3

**Summary:**

This paper considers the offline optimization problem, where offline (x, y) pairs are given, and the goal is to return samples x with better value. The paper proposes a distribution shifting approach by first learning the underlying function via a mean estimate of the Gaussian process under difference kernel functions and creating two sets of samples $X^+$ and $X^-$ by taking gradient ascent and gradient descent based on the estimated function starting from the offline samples. Then a diffusion model is learned via translating the data from $X^-$ to $X^+$ via assumed conditional Gaussian distribution. Experiments are conducted on the variance of benchmarks and the proposed method is compared to be comparable/superior to some offline linear optimization baselines.

**Strengths:**

The problem of offline optimization is important and has many potentials.

The experiments are presented clearly and the proposed method indeed has improvements compared to baselines.

**Weaknesses:**

The originality of viewing the offline optimization as a distribution shift is not novel. See [a].

The derivations are not correct. The proposed method is not theoretically justified, merely a theoretically motivated heuristic. For example, the Assumption of Eq (13) cannot be justified given Eq (12), thus giving no evidence for Eq (14) as combining both equations. Thus the overall learning procedure lacks of justifications. Take another example of Eq (3), the optimal solution x^* is a function of the unknown function g, and thus g(x^*) conditioned on the offline data is not Gaussian. It is the maximum of several Gaussian since $x^*$ is also a random variable.

The diffusion approach generates data outside of the distribution of the samples (outside of the distribution coverage) thus leading to critical generalization issues. This issue is not sufficiently addressed.

Overall, it is a work based on a wrong assumption/mathematics though has engineering improvement.

[a] Zhang, Q., Zhou, R., Shen, Y. and Liu, T., 2024. From Function to Distribution Modeling: A PAC-Generative Approach to Offline Optimization. arXiv preprint arXiv:2401.02019.

**Questions:**

In the paper, notation $\omega$ is used in multiple places. Is it a random variable indicating the randomness of the underlying function and offline data? In Eq (12) and follows, is it $\omega^*$ or $\omega$?

---

> ### Author Response · Authors · 2024-11-22
> **Response to Reviewer 7UEw -- part 1/3: Concerns and Questions 1-2**
>
> Thank you for your detailed review. We would like to your questions as below:
>
> > The originality of viewing the offline optimization as a distribution shift is not novel. See [a].
>
> We would like to emphasize that (1) viewing offline optimization from the lens of a distribution shift between the offline and high-value input regimes is a well-known perspective (see [1], [2], [3], and [4]) that predates both our work and [a]; and (2) certainly, this does not represent the novelty of our work.
>
> We also do not think any recent work in offline optimization -- including [a] -- is claiming to contribute to this well-known fact. Instead, the true novelty in existing offline optimization papers lies in their technical approaches to model and handling such distribution shifts, either empirically or theoretically.
>
> Our contribution is that we pointed out that under the mild assumption, there exists a BBDM that maps between the offline and high-value input regimes, thus casting offline optimization as learning a BBDM.
>
> On the other hand, the key contribution of [a] is learning a point-wise re-weighting function that transforms the offline input distribution into a distribution over the high-value input regime.
>
> A quick comparison with [a] on TFBIND8 also shows that our approach has an empirical advantage over [a] and DDOM [5] which [a] is based on.
>
> | Baseline  | TFBind8        |
> |-----------|---------------|
> | [a]       | 0.941 ± 0.034 |
> | DDOM      | 0.885 ± 0.061 |
> | L2HD      | 0.986 ± 0.007 |
>
>
> However, [a] presents a fantastic theoretical analysis.
>
> Given the above, the approaches in our work and [a] are different and constitute orthogonal contributions. As such, while the solution perspective in [a] is novel and interesting, it does not cancel or subsume our contribution, which is equally novel.
>
> We hope the reviewer would agree with us on this point.
>
> On the assumption front, both our work and [a] adopt different assumptions that are technically useful but not necessarily universally true (more to this later).
>
> [a] Zhang, Q., Zhou, R., Shen, Y. and Liu, T., 2024. From Function to Distribution Modeling: A PAC-Generative Approach to Offline Optimization. arXiv preprint arXiv:2401.02019.
>
> [1] Trabucco, Brandon, Aviral Kumar, Xinyang Geng, and Sergey Levine. "Conservative objective models for effective offline model-based optimization." In International Conference on Machine Learning, pp. 10358-10368. PMLR, 2021.
>
> [2] Trabucco, Brandon, Xinyang Geng, Aviral Kumar, and Sergey Levine. "Design-bench: Benchmarks for data-driven offline model-based optimization." In International Conference on Machine Learning, pp. 21658-21676. PMLR, 2022.
>
> [3] Qi, Han, Yi Su, Aviral Kumar, and Sergey Levine. "Data-driven offline decision-making via invariant representation learning." Advances in Neural Information Processing Systems 35 (2022): 13226-13237.
>
> [4] Yu, Sihyun, Sungsoo Ahn, Le Song, and Jinwoo Shin. "Roma: Robust model adaptation for offline model-based optimization." Advances in Neural Information Processing Systems 34 (2021): 4619-4631.
>
> [5] Krishnamoorthy, Siddarth, Satvik Mehul Mashkaria, and Aditya Grover. "Diffusion models for black-box optimization." In International Conference on Machine Learning, pp. 17842-17857. PMLR, 2023.
>
> > The derivations are not correct. The proposed method is not theoretically justified, merely a theoretically motivated heuristic. For example, the Assumption of Eq (13) cannot be justified given Eq (12), thus giving no evidence for Eq (14) as combining both equations. Thus the overall learning procedure lacks of justifications.
>
> We respectfully disagree with the statement that our derivations are not correct. It is correct under the assumption stated in Eq. (13) which stipulates that:
>
> (1) the distribution over the (true) optimal input can be reparameterized as a linear transformation of an isotropic Gaussian distribution; and
>
> (2) such transformation is in turn characterized via a (learnable) linear function of the output of a T-step gradient ascent using the surrogate gradient.
>
> Under this assumption, we show that there exists a BBDM that maps between the offline (low-value) and high-value input distributions.
> This is not an unreasonable assumption compared to that of [a] which stipulates that there is (1) diffusion model mapping from isotropic Gaussian to high-value input distribution, which is then (2)  pointwise transformed into the offline distribution via a (learnable) re-weighting function.
>
> I hope the reviewer will agree with us on these points.

---

> ### Author Response · Authors · 2024-11-22
> **Response to Reviewer 7UEw -- part 2/3: Concerns and Questions 3-4**
>
> >Take another example of Eq (3), the optimal solution x^* is a function of the unknown function g, and thus g(x^*) conditioned on the offline data is not Gaussian. It is the maximum of several Gaussian since x∗ is also a random variable.
>
> We believe there is a misunderstanding that $x_*$ in Eq. (3) refers to the maxima of $g(x)$. It does not.
>
> We ask the reviewer to please kindly read the sentence before Eq. (3) again which only says that $x_*$ is an unseen input.
>
> We suspect the reviewer’s confusion is caused because we previously use $x_*$ to denote the maxima of $g(x)$ in Eq. (2) but we did not mean for $x_*$ to be particularly the maximizer in the context of Eq. (3).
>
> We apologize for the overloaded notation and changed it to $x_\tau$ in the latest manuscript.
>
> The two sections 2.1 and 2.2 are separate background reviews.
>
> We assert that the use of the Gaussian process in this context is NOT to predict the value of the maximum output of $g(.)$. Instead, it is used to draw synthetic functions that are within a statistical neighborhood of the oracle function. As those sampled functions are within the statistical neighborhood of $g(.)$, their low- and high-value input regimes can be used to (approximately) represent those of the oracle, which provides training data for the BBDM.
>
> Note that using the low- and high-value points from within the offline dataset is not helpful in this case because the offline dataset is drawn from the bottom 40th (or 50th) percentile of the full oracle dataset -- see Design-bench [6]. This is why we need to use GP. A similar GP approach to generate synthetic data for pre-training offline optimizer was previously investigated in [7].
>
> [6] Trabucco, Brandon, Xinyang Geng, Aviral Kumar, and Sergey Levine. "Design-bench: Benchmarks for data-driven offline model-based optimization." In International Conference on Machine Learning, pp. 21658-21676. PMLR, 2022.
>
> [7] Nguyen, Tung, Sudhanshu Agrawal, and Aditya Grover. "Expt: Synthetic pre-training for few-shot experimental design." Advances in Neural Information Processing Systems 36 (2023): 45856-45869.
>
> >The diffusion approach generates data outside of the distribution of the samples (outside of the distribution coverage) thus leading to critical generalization issues. This issue is not sufficiently addressed.
>
> The specific data we used to train the BBDM guarantees that it will not generate data outside the distribution of the samples. To see this, note that the training data of the BBDM constitutes the low- and high-value inputs of synthetic functions which are sampled from a statistical neighborhood of the oracle. This statistical neighborhood is defined via fitting a GP posterior using the offline oracle data.
>
> The sets of low- and high-value inputs from those sampled functions therefore represent sufficiently well the low- and high-value regimes of the oracle. A BBDM learned with such data will then help map from low-value inputs to high-value inputs of the oracle.
>
> This can be verified by observing the performance of our method in the experiments. Had the BBDM produced inputs outside the high-value regime, it would have recommended low-value candidates, harming the overall performance. As our reported performance is superior to the baselines over multiple runs across various datasets, we can ascertain that our customized BBDM has not generated OOD data.
>
> In other words, the OOD issue mentioned by the reviewer has indeed been addressed in our context via the sampling of synthetic data from GP posteriors fitted on offline oracle data. This approach is well-grounded, as it has been shown in [8] that a GP with an RBF kernel is a universal approximator, capable of modeling a wide range of functions. Furthermore, prior work [9] demonstrates the feasibility of generating pseudo-labels for unlabeled designs using a vast set of GP priors, achieving impressive results. In our case, we take this further by using GP posteriors, which are specifically fitted to the offline dataset, ensuring the synthesized functions are closely aligned with the target function on the observed data.
>
> [8] Micchelli, Charles A., Yuesheng Xu, and Haizhang Zhang. "Universal Kernels." Journal of Machine Learning Research 7, no. 12 (2006).
>
> [9] Nguyen, Tung, Sudhanshu Agrawal, and Aditya Grover. "Expt: Synthetic pretraining for few-shot experimental design." Advances in Neural Information Processing Systems 36 (2023): 45856-45869.

---

> > ### Author Response · Authors · 2024-11-22
> > **Response to Reviewer 7UEw -- part 3/3: Concerns and Questions 5-6**
> >
> > > Overall, it is a work based on a wrong assumption/mathematics though has engineering improvement.
> >
> > Based on our responses to the reviewer’s previous concerns, it is now clear that our assumption in Eq. (13) is as reasonable as that of [a] -- see our response to Concerns and Questions 2-3 -- and our mathematics in Eq. (3) is not wrong.
> >
> > The perception that Eq. (3) is wrong stems from a misunderstanding that $x_*$ in the scope of Eq. (3) refers to the oracle maxima. We have explained that it does not.
> >
> > Furthermore, speaking of assumption, we would like to note that assumptions are part of model designs and to quote the famous statistician George Box:
> >
> > “All models are wrong but some are useful”
> >
> > For instance, the reviewer’s suggested work in [a] assumes that offline data points are independently drawn from some unknown distribution. However, in the DesignBench context, offline data are drawn from the bottom 40th (or 50th) percentile of the oracle dataset.
> >
> > This means this assumption/model is wrong in the DesignBench context but we believe it is useful in the broader context where an interesting theoretical analysis has been established.
> >
> > Likewise, it is also unclear whether our assumption in Eq. (13) is universally true but observing the reported performance, we know that it is useful.
> >
> > > In the paper, notation ω is used in multiple places. Is it a random variable indicating the randomness of the underlying function and offline data? In Eq (12) and follows, is it ω∗ or ω?
> >
> > As stated in the Problem Definition section, $\omega$ ($\omega_*$) represents the parameter of the surrogate model $g(x; \omega)$ trained to approximate this black-box function $g(x)$.
> >
> > Overall, we hope our response has sufficiently addressed the reviewer’s questions. We hope the reviewer will consider increasing the overall rating of our work.
> >
> > We thank the reviewer for pointing out the interesting work [a], which we will cite and discuss in our related work section.

---

> > > ### Author Response · Authors · 2024-11-23
> > > **Response to Reviewer 7UEw -- Follow-up**
> > >
> > > Dear Reviewer **7UEw**,
> > >
> > > It seems that there might have been a few notational misunderstanding that lead to your assessment that our work is based on wrong assumptions and mathematics.
> > >
> > > We have clarified that this is not the case.
> > >
> > > We have also read the suggested work [a] (which is very interesting) and as we explained in our first response above:
> > >
> > > 1. The novelty of both our work and [a] does not lie in the recognition of offline optimization as a distributional gap challenge. That insight long predates both these works.
> > >
> > > 2. The novelty of our work and [a] lies in how we characterize and compensate for such distributional gap. We have clearly articulated the specific novelty of both works (see our first message) and we believe our work and [a] are orthogonally novel.
> > >
> > > Given the above, we hope the reviewer would reconsider the assessment of our work.

---

> > > > ### Comment · Reviewer_7UEw · 2024-11-24
> > > >
> > > > I agree with that “All models are wrong but some are useful”. “Wrong” means the model may not capture the reality, since the mathematical model of reality is too complicated to have a clean model. The model and its follow-up derivations should still be mathematically correct. The unknown function is modeled via Gaussian process prior, which is acceptable even though “wrong” in the sense of may not reflecting the reality, however equations Eq (12) and Eq (13) are mathematically contradicted to each other, under the GP model.

---

> > > > > ### Author Response · Authors · 2024-11-25
> > > > > **There is no contradiction between GP & Eq. (12) & (13).**
> > > > >
> > > > > Dear Reviewer 7UEw,
> > > > >
> > > > > There is no contradiction between the GP **output model** — in Eq. (3) — and the high-value **input model** in Eq. (12) & (13).
> > > > >
> > > > > We understand that the output of the maxima of a GP-distributed function is not Gaussian as you had mentioned in your original review.
> > > > >
> > > > > > [Take another example of Eq (3), the optimal solution x^* is a function of the unknown function g, and thus g(x^*) conditioned on the offline data is not Gaussian. It is the maximum of several Gaussian since x∗ is also a random variable.]
> > > > >
> > > > > **But, this has never been a part of our model.** As we pointed out in our earlier response, the notation $x_*$ in Eq. (3) does not refer to the maxima in Eq. (2). This was a notation overload that we had corrected in our uploaded revision - **Eq. (3) is therefore correct**
> > > > >
> > > > > In fact, the GP assumption is set on the output distribution while Eq. (12) & (13) are set on the high-value input distribution. These are orthogonal and do not contradict each other as further elaborated below.
> > > > >
> > > > > 1. Eq. (13) assumes that the **distribution over a regime of high-value inputs** can be re-parameterized as a distributional transformation of an isotropic Gaussian distribution where the transformation is based on the output of a T-step surrogate-based gradient ascent of a neural surrogate in Eq. (12). This means there exists a BBDM mapping from the distribution of offline inputs to the distribution of high-value output
> > > > >
> > > > > 2. To learn this BBDM model, we need to sample from the high-value input regime of the oracle which is not accessible to us. Hence, we sidestep this by sampling instead from the high-value regimes of similar functions of the oracle. These similar functions are GP posteriors (with different kernel initializations) fitted on the oracle’s offline data. **Again, these similar functions are NOT conditioned on the maxima output of the oracle so the GP mathematics in Eq. (3) is NOT wrong.**
> > > > >
> > > > > Please let us know if the above has addressed your concern.
> > > > >
> > > > > Otherwise, could you please kindly elaborate on which parts of our models that are contradicting each other?
> > > > >
> > > > > Thank you for your response and we look forward to hearing back from you soon!

---

### Official Review · Reviewer_c7rs · 2024-11-05

**Soundness:** 4
**Presentation:** 2
**Contribution:** 3
**Rating:** 6
**Confidence:** 3

**Summary:**

This is a high-potential paper which could be much better written. I suggest a full review of the paper for readability. For example, the second line of the abstract is 4 lines long.  The goal of the paper is not clear from reading the introduction.

The paper seeks to learn surrogate parameterized functions which learn the data distribution. Using these surrogates, the data is augmented, and this augmented data can then be used to train a model. The issue with such augmentation is that both the (x,y) pairs must be learnt, and predicting the x may lead to OOD generations. This is usually solved through inverse mappings, or search. This paper proposes ideas from Brownian bridge diffusion, which was previously proposed in computer vision to extend to tasks in the Design Bench.

**Strengths:**

- The use of diffusion in design tasks is quite novel.
- The results are strong. But note that the best of 128 samples is reported in Table 1. The algorithm itself may not have access to "best score". So while the algorithm reaches good results in best case, median or avg may have to be considered.

**Weaknesses:**

- Why use the brownian bridge diffusion process? How do other diffusion processes perform?
- Section 3.1 what is the stochastic translation operator? why use terms before defining them?
- The paper has spent a lot of space on Section 3.1 which comes without explanation of what the goal is. For example what is g in this setting? Are these the equivalents of the noise estimates in DDPM? What is the reason for the specific form in equation 13? Please rewrite thsi section with what you are attempting in this section. While reading this section, it is extremely unclear what the mathematics is leading towards, and hence why a stochastic translation is required.
- There is no algorithm section translating the mathematics of Section 3 into the experiments of Section 4. I do not see what was the optimization implemented, or how it was obtained.
- The paper does not report running times of the algorithms. Given diffusion can be significantly costlier it is worth reporting the wall-clock time.
- The paper reports max score in Table 1, but without external evaluation, it is not clear that the algorithm may know what is best. Hence this is not the expected performance. Can  you also show the mean score?

**Questions:**

- Why do you set the value of zeta to t/T? and delta_t to zeta_t(1-zeta_t)?

### Suggestions:

- Could you consider revising your abstract? The goal of the paper was not immediately clear.
- Please consider a rewrite of section 3.1. Start with what the optimization function is. Then follow through with the derivations on how it was obtained. Please mention how this is used in an algorithm.
- Happy to raise my score given some changes to the manuscript.

### Typos
- Line 233: "as follows"

---

> ### Author Response · Authors · 2024-11-22
> **Response to Reviewer c7rs -- part 1/2: Concerns and Questions 1-5**
>
> Thank you very much for your positive feedback and detailed suggestions. We have addressed your concerns as follows:
>
> >This is a high-potential paper which could be much better written. I suggest a full review of the paper for readability. For example, the second line of the abstract is 4 lines long. The goal of the paper is not clear from reading the introduction.
>
> To improve the clarity of the abstract and introduction, we have revised both sections with highlighted changes to clearly articulate the paper's objectives. For these updates, please refer to the latest version of our manuscript, which has been recently uploaded.
>
> > The results are strong. But note that the best of 128 samples is reported in Table 1. The algorithm itself may not have access to "best score". So while the algorithm reaches good results in best case, median or avg may have to be considered.
>
> We would like to point out that in addition to the best case (100th percentile) result in Table 1, we have also provided results for the median (50th percentile) and 80th percentile in Tables 8 and 9 in Appendix B, which are referred to in lines 363-364 in the main text. These tables demonstrate that our method, L2HD, achieved state-of-the-art performance not only in the 100th percentile but also in the 80th and 50th percentiles with the highest rank. Furthermore, we have included additional violin plots in Appendix B.3 and Figure 2, illustrating that L2HD has a score distribution skewed towards higher-value regions compared to the other baselines.
>
> > Why use the brownian bridge diffusion process? How do other diffusion processes perform?
>
> Our work has reformulated offline optimization as a distributional translation task, where the goal is to map a low-value distribution to a high-value distribution. This allows us to draw from powerful methods in the image-to-image translation community, and BBDM [1] is one such approach. In fact, we have shown in Section 3.1 that under a certain modeling assumption and parameterization, there exists a BBDM that maps between those distributions. Hence, BBDM is our natural choice. However, as far as a practical solution goes, any diffusion model capable of translating between two (implicit) distributions (i.e., those that are represented implicitly with a set of samples) could also be applied to our framework. Exploring potential uses of other diffusion processes is an interesting direction to extend our current work but it is orthogonal to the current scope. We will consider this in a potential follow-up of our work. Thank you very much for your suggestion.
>
> [1] Li, Bo, Kaitao Xue, Bin Liu, and Yu-Kun Lai. "Bbdm: Image-to-image translation with brownian bridge diffusion models." In Proceedings of the IEEE/CVF conference on computer vision and pattern Recognition, pp. 1952-1961. 2023.
>
> > Section 3.1 what is the stochastic translation operator? why use terms before defining them?
>
> The parameterized scaling and stochastic translation operator, $(\zeta_t, \delta_t)$, refers to the operations in Eq. (13) (Eq. (15) in our latest manuscript), where we scale the gradient sum by $\zeta_t$ and add Gaussian noise (translation) scaled by $\delta_t$​. To improve clarity and avoid confusion, we have revised the terms in our revised manuscript (from Eq. (13) to Eq. (15)).
>
> > The paper has spent a lot of space on Section 3.1 which comes without explanation of what the goal is. For example what is g in this setting? Are these the equivalents of the noise estimates in DDPM?
>
> We would like to clarify that the goal of Section 3.1 is stated in lines 208-214 which is at the beginning of Section 3.1. We believe these lines provide sufficient context for the detailed content in the section, outlining the objectives and rationale behind the methodology discussed.
>
> As stated in lines 101-105 in the Problem Definition section, $g(x)$ represents the black-box function, and $g(x; \omega)$ denotes the surrogate model trained to approximate this black-box function $g(x)$. Hence, they are not the equivalents of the noise estimates in DDPM.

---

> > ### Author Response · Authors · 2024-11-22
> > **Response to Reviewer c7rs -- part 2/2: Concerns and Questions 6-10**
> >
> > > What is the reason for the specific form in equation 13? Please rewrite thsi section with what you are attempting in this section. While reading this section, it is extremely unclear what the mathematics is leading towards, and hence why a stochastic translation is required.
> >
> > We have revised Section 3.1 to enhance the clarity of our work. For the reviewer’s convenience, we summarize it below:
> >
> > Our work aims to optimize a black-box function using only an offline dataset of its observed input and output -- see Eq. (1) and Eq. (2). The vanilla approach in (2) is problematic due to the distribution shift between the offline (low-value) and high-value input regimes of the oracle.
> >
> > To handle this distribution shift we assume in Eq. (13) that:
> >
> > (1) the distribution over the (true) optimal input can be reparameterized as a linear transformation of an isotropic Gaussian distribution; and
> >
> > (2) such transformation is in turn characterized via a (learnable) linear function of the output of a T-step gradient ascent using the surrogate gradient.
> >
> > Under this assumption, we show that there exists a BBDM that maps between the offline (low-value) and high-value input distributions of the oracle -- see Eq. (14) - Eq. (17).
> >
> > To learn this BBDM, we sampled synthetic functions from a statistical neighborhood of the oracle. This statistical neighborhood is defined via fitting GP posteriors using the offline oracle data -- see Section 3.2.
> >
> > The sets of low- and high-value inputs from those sampled functions -- see Eq. (19) and Eq. (20) -- therefore represent sufficiently well the low- and high-value regimes of the oracle. A BBDM learned with such data will then help map from low-value inputs to high-value inputs of the oracle.
> >
> > > There is no algorithm section translating the mathematics of Section 3 into the experiments of Section 4. I do not see what was the optimization implemented, or how it was obtained.
> >
> > We would like to point out that a detailed algorithm pseudocode translating the mathematics of Section 3 into implementation has been provided in Appendix C. Please see Algorithms 1 and 2.
> >
> > >The paper does not report running times of the algorithms. Given diffusion can be significantly costlier it is worth reporting the wall-clock time.
> >
> > Thank you for the suggestion. We reported the incurred wall-clock time of all baselines below.
> >
> > | Model         | Ant   | Dkitty | TFBind8 | TFBind10 |
> > |---------------|-------|--------|---------|----------|
> > | GA            | 122s  | 200s   | 124s    | 420s     |
> > | BO-QEI        | 219s  | 310s   | 402s    | 383s     |
> > | CMA-ES        | 5747s | 4889s  | 2667s   | 4323s    |
> > | MINS          | 797s  | 998s   | 2213s   | 792s     |
> > | REINFORCE     | 240s  | 257s   | 507s    | 373s     |
> > | CBAS          | 394s  | 390s   | 835s    | 529s     |
> > | ICT           | 188s  | 189s   | 216s    | 639s     |
> > | Tri-mentoring | 4276s | 3921s  | 4673s   | 3410s    |
> > | **L2HD**          | 298s  | 297s   | 407s    | 575s     |
> >
> > The reported running time shows that even with the GP fitting overhead, L2HD still runs faster than several baselines such as CMA-ES, MINS, and Tri-mentoring.
> >
> > We believe this addresses the concern and demonstrates that the diffusion model's computational cost is manageable and well-suited to the current scope of our work.
> >
> > > The paper reports max score in Table 1, but without external evaluation, it is not clear that the algorithm may know what is best. Hence this is not the expected performance. Can you also show the mean score?
> >
> > Yes. We reported results for the median (50th percentile) and 80th percentile in Tables 8 and 9 in Appendix B, which are referred to in lines 363-364 in the main text. These tables demonstrate that our method, L2HD, achieved state-of-the-art performance not only in the 100th percentile but also in the 80th and 50th percentiles with the highest rank. Furthermore, we have included additional violin plots in Appendix B.3 and Figure 2, illustrating that L2HD has a score distribution skewed towards higher-value regions compared to the other baselines.
> >
> > > Why do you set the value of zeta to t/T? and delta_t to zeta_t(1-zeta_t)?
> >
> > We can choose arbitrary values for $\zeta_t$ and $\delta_t$. Interestingly, we found that by setting $\zeta_t = 1-m_t =1- t/T$ and $\delta_t = 2(m_t - m_t^2)$, the process becomes a Brownian Bridge process, as described in the paper BBDM. This formulation allows us to leverage the strengths of BBDM to address the translation problem effectively.
> >
> >
> > Overall, we have revised both the abstract and introduction, as well as rewritten Section 3.1 following your suggestions. We hope that our answer and revision have addressed your concern.
> >
> > If so, we hope you will consider increasing the overall rating of our work. Otherwise, please let us know if you have any follow-up questions for us.

---

> > > ### Author Response · Authors · 2024-11-24
> > > **Response to Reviewer c7rs -- Follow-up**
> > >
> > > Dear Reviewer c7rs,
> > >
> > > We have revised our abstract, introduction, and Section 3.1 following your suggestions.
> > >
> > > May we know if our revision has made our contribution clearer?
> > >
> > > We would appreciate your feedback on this as well as our specific response to each of your questions.
> > >
> > > Thank you for your consideration.
> > >
> > > Best regards,
> > >
> > > Authors

---

> ### Author Response · Authors · 2024-11-29
> **Response to Reviewer c7rs -- Follow-up**
>
> Dear Reviewer **c7rs**,
>
> To assist with your final assessment of our paper, we provide the following summary of our work:
>
> **Motivation:**
>
>   * Black-box optimization fundamentally seeks **a sequence of iterates $x_t$ that map from a low-value design ($x_T$) to a high-value design ($x_0$)**. A naive approach uses **a surrogate-based gradient ascent (as shown in Eq. (12))**, relying solely on offline data (e.g., oracle data from low-value regimes). However, this approach often **hallucinates mapping targets instead of defining them properly.**
>
> **Idea:**
>
>   * To address this limitation, we assume that under the GP-based stochasticity of the oracle, the **distribution of high-value designs can be derived from a transformation of isotropic Gaussian noise (Eq. (13))**. This assumption imposes a constraint on the sequence of iterates, **defining how the mapping target influences the dynamics (Eq. (15)).**
>
> Instead of learning Eq. (12) for offline optimization, **we propose learning a transition dynamic that satisfies Eq. (15)**, which characterizes a family of **Brownian bridge diffusion processes.**
>
> **Technical Challenges:**
>
>   * Learning the transition dynamics to generate iterates satisfying Eq. (15) is challenging, as it **requires observing high-value samples from the oracle, which are unavailable.**
>
> However, we empirically find that **high-value samples from functions similar to the oracle can also be used as effective surrogates to represent the oracle's high-value regime.** Sampling such functions is feasible using the GP posterior distribution of the oracle.
>
> **Results:**
>
>   * Our approach significantly outperforms the state-of-the-art methods, demonstrating its efficacy in offline black-box optimization.
>
> —
>
> **Further Insights:**
>   * Overall, our approach features **a new connection between offline optimization and the Brownian bridge diffusion process which has not been previously explored.** We believe this is a very new perspective that would be useful for future research development in offline optimization.

---

> > ### Author Response · Authors · 2024-12-01
> > **Response to Reviewer c7rs -- Follow-up**
> >
> > Dear Reviewer c7rs,
> >
> > As we are approaching the end of the rebuttal period, could you please kindly share with us your thoughts regarding the improvement we have made to the manuscript following your suggestions?
> >
> > We have also provided a concise summary of our work in the previous message.
> >
> > Thank you for your consideration.

---

> > > ### Author Response · Authors · 2024-12-03
> > > **Response to Reviewer c7rs -- Follow-up**
> > >
> > > Dear Reviewer c7rs,
> > >
> > > As the discussion period is closing soon, we sincerely hope you could kindly take a look at our revision and let us know if it tells a clearer story now. We have put significant effort into revising the paper and carefully addressing all the feedback provided. Acknowledgment of these efforts and a potential adjustment in the score, if deemed appropriate, would mean a great deal to us.
> > >
> > > Thank you for your time and consideration.

---

### Author Response · Authors · 2024-12-03
**Comprehensive Resolution of All Post-Rebuttal Concerns.**

Dear Area Chair and Reviewers,

We would like to express our sincere gratitude to the Area Chair for coordinating the review process and to the reviewers for their detailed reviews.

1. We are particularly thankful to Reviewer **3WMn** for the acceptance score.

2. We are thankful that Reviewer **c7rs** gave us excellent and good ratings for soundness and contributions, respectively. We believe the reviewer’s overall rating of 5 is mainly regarding the presentation, which has been thoroughly fixed following the detailed feedback from the reviewer. We still hope the reviewer would be able to confirm if our latest revision (uploaded to OpenReview) is sufficient before the author-reviewer discussion closes. We have put in a lot of effort to revise the paper and your feedback on the revision would mean a lot to us.

3. We also appreciate Reviewer **7UEw**’s critical review of our paper. Given the discussion, we understand that the single remaining concern of the reviewer is that our assumption in Eq. (13) is self-contradicting due to the misunderstanding that the derivative of a GP-distributed function might not be distributed by a GP, which would invalidate the Gaussian assumption in Eq. (13). We clarified that this is indeed a misunderstand as the derivative of a GP-distributed function is also distributed by a GP, which is explicitly stated in the first sentence of Section 9.4 of the Gaussian process for Machine Learning (GPML) textbook by Carl Edward Rasmussen and Chris Williams.

    We have also provided a mathematical prove that there is no contradiction in Eq. (13) using this fact — please see our second to last message to Reviewer **7UEw**.

4. We also emphasize that our assumption is equivalent to assuming that the mapping between the oracle’s low- and high-value input distributions follows a Brownian bridge diffusion process. This is a single assumption and if it were indeed self-contradictory, it would imply a fundamental flaw in the concept of the Brownian bridge diffusion process which is implausible.

     In addition, we hope Reviewer **7UEw** will consider give credit to our significant empirical results. We believe this result is solid and the use of Brownian bridge diffusion in offline optimization is truly the first of its kind. Usually, a model based on conflicting assumption will not perform well across multiple datasets. This is certainly not the case here.

---

## Summary of Responses

1. **Clarity Improvements:**
   - Revised the abstract and introduction to better articulate the paper’s objectives, with highlighted changes for transparency.
   - Improved Section 3.1 to enhance the clarity of our approach. *(c7rs)*

2. **Performance Comparison:**
   - Reported wall-clock times for all baselines, demonstrating that L2HD outperforms several baselines, including CMA-ES, MINS, and Tri-mentoring. *(c7rs, 3WMn)*

3. **Addressing Novelty and Assumption Concerns:**
   - Clarified the novelty of our contributions and established mathematically that there is no contradiction in our assumption. *(7UEw)*

---

Overall, based on the above, we strongly believe that our rebuttal has effectively addressed all concerns raised.

We sincerely thank the Area Chair and reviewers for their time and effort in evaluating our paper. We hope the final assessment reflects the strengths and contributions of our work.

Best regards,

The Authors

---

### Meta-Review · Area_Chair_uumY · 2024-12-21

**Metareview:**

This paper introduces a method for offline black box optimization, where it can be challenging to learn a surrogate function that generalizes to out-of-distribution designs. The authors propose using a Brownian bridge diffusion model to transform "low-valued" data (i.e., input/output pairs where the output is suboptimal) into "high-valued" data (input/output pairs where the output is optimal).

The approach proposed by this paper is novel and demonstrates positive results. The primary issue with this paper is in the presentation. The goal of the paper is not immediately clear from the abstract and introduction; the methods section was initially confusing, and one reviewer had trouble determining if the method was valid. While the authors made an effort to improve the presentation during the revision period, this paper would benefit from another round of reviewing. Given that most reviewers cited presentation issues in their review, I hope that the authors put in additional effort towards improving the clarity of their manuscript, as focusing on the writing will ultimately lead to a publication with a higher impact.

**Additional Comments On Reviewer Discussion:**

Two of the reviewers were relatively silent during the author/reviewer discussion period. Both reviewers brought up presentation issues as well as minor technical concerns (e.g., runtime, a table for mean score, etc.). The authors responded to these technical concerns and also revised some text to improve the presentation.

One reviewer had a lengthy back-and-forward with the authors on what they perceived to be faulty modelling assumptions. Ultimately, the authors were able to demonstrate that the reviewer was mistaken. Though the reviewer did not acknowledge their mistake in the public comments, we went through the issue during the AC/reviewer discussion period and came to the correct understanding. Nevertheless, the reviewer had concerns about the presentation, which I was inclined to side with, given that these issues were shared amongst all reviewers.

---

### Decision · Program_Chairs · 2025-01-22

Reject